# GRAPHON CROSS-VALIDATION: ASSESSING MODELS ON NETWORK DATA

**Huimin Cheng**[*]
Department of Biostatistics
Boston University
Boston, MA, USA
`huimin23@bu.edu`

**Yongkai Chen**[*]
Department of Statistics
Harvard University
Cambridge, MA, USA
`yongkaichen@fas.harvard.edu`

**Ping Ma**
Department of Statistics
University of Georgia
Athens, GA, USA
`pingma@uga.edu`

**Wenxuan Zhong**[†]
Department of Statistics
University of Georgia
Athens, GA, USA
`wenxuan@uga.edu`

## ABSTRACT

Graphon models have emerged as powerful tools for modeling complex network structures by capturing connection probabilities among nodes. A key challenge in their application lies in accurately characterizing the graphon function, particularly with respect to parameters that govern its smoothness, which significantly impact the estimation accuracy. In this article, we propose a novel graphon cross-validation method for selecting tuning parameters and estimation approaches. Our method is both theoretically sound and computationally efficient. We show that our proposed cross-validation score is asymptotically parallel to the estimation error, and the selected model asymptotically converges to the optimal model. Through extensive simulations and real-world applications, we demonstrate that our method consistently delivers superior computational efficiency and accuracy.

## 1 INTRODUCTION

Graphon models represent cutting-edge statistical and machine-learning techniques for describing intricate network structures. In these models, the presence of an edge between nodes $i$ and $j$ hinges on a probability, independently determined by a measurable symmetric graphon function $f$ across the unit square (Lovász and Szegedy, 2006). The graphon function $f$, the key to the model, gauges the probability of node connections. Accurately estimating $f$ poses a significant challenge to the models' general application. The efficacy of existing estimation methods is intrinsically linked to the precise calibration of tuning parameters. For instance, the accuracy of the graphon function estimated using the sort-and-smoothing (SAS) method (Chan and Airoldi, 2014) or the neighborhood-smoothing (NS) method (Zhang et al., 2017) is heavily contingent on the chosen size of the neighborhood. This parameter is analogous to the bandwidth in kernel smoothing and is crucial for the estimation process. As depicted in Figure 1, varying the neighborhood size parameter can result in markedly different estimates of the graphon function, highlighting the sensitivity of the model to this tuning parameter.

Cross-validation (CV) is widely regarded as the gold standard for parameter tuning and model selection by dividing the dataset into subsets, training the model on a portion of the data, and then testing it on the remaining data. The primary goal of cross-validation is to assess how well a model generalizes to new, unseen data. Yet, when applied to network data, traditional CV encounters a fundamental obstacle: it assumes the independence of observations—a condition that is invariably violated by the interconnected nature of network nodes. Consequently, the standard practice of random sampling on nodes falls short, calling for an alternative approach to data splitting, which is critical to the integrity of the model validation process.

---

[*]Equal contribution.
[†]Corresponding author.

Figure 1: From left to right, we present the heatmap of true graphon probability matrix, the estimated graphon probability matrix (under low, medium, and large hyperparameters).

In graphon models, where edges are presumed to follow an independent Bernoulli distribution, the method of edge sampling takes on particular importance compared to node sampling. Common practices involve randomly removing edges and using the remaining graph for training. However, direct edge sampling can change the network's inherent topology and connectivity, thereby altering neighborhood structure, which introduces a significant bias in the estimation of the graphon function. In addition, direct edge sampling may lead to a sampling bias when the network is partitioned into training and testing sets, potentially resulting in different distributions. Such bias has profound implications: it may trigger an inaccurate estimation of the standard deviation, distort confidence intervals, and undermine hypothesis testing reliant on exact margins of error. Ultimately, it may elevate prediction error, thereby impairing the model's predictive efficacy. In response, Li et al. (2020a) proposes imputing entries of matrix $\mathbf{A}$ corresponding to reserved node pairs using a matrix completion algorithm. However, a critical condition of this approach is that $\mathbf{P}$ must be a low-rank matrix, which can be violated in dense networks. Furthermore, while theoretical guarantees exist for specific models like the Stochastic Block Model and the Random Dot Product Graph Model within the graphon model class, they are lacking for the broader category. Additionally, the computational cost of iteratively conducting matrix completion is prohibitively expensive, hindering the general applicability of the proposed algorithm in practical settings.

To face these challenges, this paper introduces a perturbation-based sampling method that innovatively splits data into training and testing sets. This method judiciously applies controlled perturbations to the network edges, thereby ensuring that training and validation data remain representative of the network's comprehensive distribution. Our condition is that the graphon function's smoothness is not compromised by such perturbations, allowing for a more faithful estimation of the true graphon. The method is versatile, accommodating varying degrees of sparsity and complexity within the network. We rigorously validate our approach against standard edge sampling methods, employing a series of experiments that underscore both its theoretical and practical merit.

## 2 MODEL SETUP

Network analysis was originally developed to map out the relationships or connections between various entities, which can represent either samples or features. A network $G = (\mathcal{V}, \mathcal{E})$ is composed of a vertex set $\mathcal{V} = \{v_i | i = 1, 2, \ldots, n\}$ and an edge set $\mathcal{E} = \{(v_i, v_j) | v_i, v_j$ are connected$\}$. Let $I_{\{\cdot\}}$ be an indicator function that maps subset $\{\cdot\}$ to one and all other elements to zero. Mathematically a network can be represented by a $n \times n$ binary matrix, say $\mathbf{A}$, with the $(i, j)$th entry $a_{ij}$, where $a_{ij} = I_{\{(v_i, v_j) \in \mathcal{E}\}}$. This matrix $\mathbf{A}$ is generally referred to as the adjacency matrix.

In this article, we assume that the $a_{ij}$s are independent and follow Bernoulli distributions with mean $p_{ij}$ respectively, i.e.,

$$a_{ij} \stackrel{\text{ind}}{\sim} \text{Ber}(p_{ij}), \tag{1}$$

Model (1) is known as the inhomogeneous random graph model (Söderberg, 2002; van der Hofstad, 2013; Ghoshdastidar et al., 2020). Clearly, $p_{ij}$ is not estimable unless we impose a specific constraint. Consequently, we further assume:

$$p_{ij} = f(\mu_i, \mu_j), \tag{2}$$

where $f : [0, 1] \times [0, 1] \to [0, 1]$ is referred to as the graphon function and the function is smooth. Here, $\mu_i$ and $\mu_j$ represent latent parameters associated with vertices $v_i$ and $v_j$, respectively. Without loss of generality, we further assume that $f$ is a bounded symmetric function, and additionally, $\mu_i$ for $1 \leq i \leq n$ are independently and identically distributed from a uniform distribution on interval $[0, 1]$.

Clearly, $f$ is neither unique nor identifiable as $f$ and $\mu_i$s are confounding with each other (Diaconis and Janson, 2007). Thus, instead of estimating $f$, researchers usually focus on the estimation of the $n \times n$ probability matrix $\mathbf{P} = [p_{ij}]$. The combined model described by (1) and (2) is commonly referred to as the graphon model. Graphon model is a general non-parametric model (Chan and Airoldi, 2014) which encompasses a variety of model classes, each with its own conditions on $f$. Common model classes include piecewise constant graphons, piecewise linear graphons, and smooth graphons. Model selection involves choosing the appropriate class that best captures the underlying structure of the graph data.

Given $f$'s inherent smoothness, researchers commonly embrace the concept of neighborhood smoothing methods, often utilized in spatial analysis, to gauge $p_{ij}$ by capitalizing on adjacent node values. Noteworthy methodologies in this domain encompass the neighborhood smoothing (NS) method (Zhang et al., 2017), which amalgamates nodes sharing common neighbors, and the sort-and-smoothing (SAS) method (Chan and Airoldi, 2014), which pool nodes with akin degrees. Both techniques center on identifying neighboring nodes within a network, requiring the establishment of a neighbor. The parameters that define the neighbor usually gauge the accuracy of the estimates.

## 3 GRAPHON CROSS-VALIDATION WITH RANDOM IMPUTATION

Let $\hat{\mathbf{P}}(M|\mathbf{A})$ be the estimated probability matrix of observing $\mathbf{A}$ given model $M \in \mathcal{M}$, where $\mathcal{M}$ represents the candidate pool of models. The performance of the model $M$ can be assessed via the mean squared error

$$L(M) = \frac{1}{n(n-1)}||\hat{\mathbf{P}}(M|\mathbf{A}) - \mathbf{P}||_F^2. \tag{3}$$

Given $\mathcal{M}$, the model $M_o \in \mathcal{M}$ that minimizes $L(M)$ represents the ideal choice one would like to make given the observed $\mathbf{A}$ and is referred to as the optimal model. Given the need for edge sampling methods that preserve the network's structural integrity while minimizing the introduction of biases, we propose the $K-$fold cross-validation with a random imputation method. The general idea of this approach is to treat the edges in the validation set as missing values that need to be randomly imputed before model training.

We randomly partition the set $\{(v_i, v_j) \in \mathcal{V} \times \mathcal{V} : i < j\}$ into K subsets $\mathcal{S}_1, \cdots, \mathcal{S}_K$, where each subset contains approximately $\frac{n(n-1)}{2K}$ node pairs. This partitioning ensures that the subsets are of approximately equal size and that each node pair has an equal chance of being included in any of the subsets. Let $\mathbf{A}^{[k]}$ denote the set of observed entries $a_{ij}$ for node pairs within $\mathcal{S}_k$. For the $k$th fold validation, we generate the $n \times n$ training adjacency matrix as $\mathbf{A}^{[-k]}$, of which the $ij$th entry is

$$\mathbf{A}_{ij}^{[-k]} = \begin{cases} a_{ij} & \text{if } (v_i, v_j) \notin \mathcal{S}_k \\ b_{ij} & \text{otherwise,} \end{cases} \tag{4}$$

where $b_{ij}$s are independently identically sampled from the Bernoulli distribution with mean $\theta$. This equation defines how the training adjacency matrix is constructed by incorporating the observed training entries $a_{ij}$ and randomly imputed entries $b_{ij}$. $\theta$ serves as a tuning parameter and remains fixed as a constant throughout our procedure. The selection of $\theta$ is discussed in Section S.4.

We have the following Lemma:

**Lemma 1.** *If $\mathbf{A}$ follows a graphon model with $E(\mathbf{A}) = \mathbf{P}$, we can deduce that the value of any entry in $\mathbf{A}^{[-k]}$ is mutually independent of the node connectivity for node pairs in the validation set $\mathcal{S}_k$. More precisely, given $\mathbf{P}$, we have that $\mathbf{A}_{ij}^{[-k]}$ is distributed according to a Bernoulli distribution with parameter $w_k\theta + (1 - w_k)p_{ij}$, where $w_k$ represents the proportion of number of node pairs in $\mathcal{S}_k$ and $p_{ij}$ represents the linking probability for $(v_i, v_j)$.*

Lemma 1 ensures the independence between $\mathbf{A}^{[-k]}$ and $\mathbf{A}^{[k]}$ given $\mathbf{P}$, which follows directly from the independence of the edges. It also demonstrates that despite the training set and the original graph holding different distributions, their distribution can be mapped to each other through an affine transformation. Let $\mathbf{P}^{[-k]}$ denote the nodes connecting probability matrix of the $n \times n$ adjacency matrix $\mathbf{A}^{[-k]}$. We have

$$\mathbf{P}^{[-k]} = w_k\theta\mathbf{1}\mathbf{1}^T + (1 - w_k)\mathbf{P}, \tag{5}$$

where $\mathbf{1}$ is the ones vector. The proof of Lemma 1 is provided in the Appendix.

Equation (5) implies that with the kth fold training data $\mathbf{A}^{[-k]}$ and model $M$, we can obtain a predictor of $\mathbf{P}$ as follows:

$$\hat{\mathbf{P}}_k(M) = \frac{\hat{\mathbf{P}}(M|\mathbf{A}^{[-k]}) - w_k\theta\mathbf{1}\mathbf{1}^T}{1 - w_k},\tag{6}$$

where $\hat{\mathbf{P}}(M|\mathbf{A}^{[-k]})$ is the estimate of $\mathbf{P}^{[-k]}$ computed based on method $M$ or model $M$ with the training data $\mathbf{A}^{[-k]}$. Let $\hat{p}_{ij}^{[k]}(M)$ denote the predicted probability for validation node pairs $(v_i, v_j) \in \mathcal{S}_k$. It is evident that $\hat{p}_{ij}^{[k]}(M)$ corresponds to the $(i, j)$th entry of $\hat{\mathbf{P}}_k(M)$. Therefore, $M$ can be selected if it minimizes the following prediction error:

$$V_K(M) = \frac{2}{n(n-1)} \sum_{k=1}^{K} \sum_{(v_i, v_j) \in \mathcal{S}_k} (\hat{p}_{ij}^{[k]}(M) - a_{ij})^2.\tag{7}$$

In practice, $\hat{p}_{ij}^{[k]}(M)$ may exceed one or fall below zero due to computational or measurement errors. In such cases, we truncate the estimated $\hat{p}_{ij}^{[k]}(M)$ to ensure it lies within the interval $[0, 1]$. Given a set of candidate models $\mathcal{M}$, we select the optimal model $M_V$ which minimizes $V_K(M)$. The detailed algorithm for the proposed K-fold cross-validation is provided in the Appendix Algorithm 1.

**Computational cost.** To evaluate a set of $|\mathcal{M}|$ candidate hyperparameters on an $n$-node network using $K$-fold cross-validation, the total computational complexity of our CV-imputation method is $O(|\mathcal{M}| \cdot (KC_{estim}(n) + n^2))$, where $C_{\text{estim}}(n)$ denotes the cost of the specific graphon estimator (e.g., NS, ICE). The detailed analysis can be found in Section S.8. Existing graphon estimation method usually have $C_{estim}$ greater than $n^2$. So the cost is mainly dominated by the graphon estimation method itself rather than by CV-imputation itself. In contrast, the competing ECV method has complexity $O(|\mathcal{M}| \cdot (KC_{estim}(n) + KT_{\text{mc}}(n)))$ where $T_{\text{mc}}(n)$ is the matrix completion complexity. The critical difference between our CV-imputation and ECV is the additional overhead per fold: Our method adds a simple $O(n^2)$ cost for imputation and transformation. ECV adds a much larger $O(T_{mc}(n))$ cost for matrix completion, which is typically $O(n^3)$ for a full SVD. Thus our method is more computationally efficient than ECV. For very large networks, a practical way to scale CV-imputation is to combine it with network subsampling: (1) extract a structurally representative subgraph using network sampling methods (e.g., Metropolis-Hastings random walk Hu and Lau (2013), curvature-based sampling Wu et al. (2023)), (2) apply CV-imputation on this subgraph to tune hyperparameters, and (3) fit the full network using the selected hyperparameter.

## 4 THEORETICAL JUSTIFICATION

A set of potential graphon estimates is denoted by $\hat{\mathbf{P}}(M|\mathbf{A})$, where $M \in \mathcal{M}$. The CV-imputation score is crafted to fine-tune $M$ to select the best estimate from this set. In this section, we offer asymptotic justifications for the CV-imputation score.

We aim to demonstrate that $V_K(M)$ approximates $L(M)$ uniformly up to a constant by postulating assumptions on the maximum $K$-fold optimism bias:

$$Q_K(M) = \sup_{1 \leq k \leq K} \frac{1}{n(n-1)} \left\| \hat{\mathbf{P}}(M|\mathbf{A}) - \hat{\mathbf{P}}_k(M) \right\|_F^2,$$

where $\|\cdot\|_F$ denotes the Frobenius norm. The optimism bias quantifies the maximum prediction bias between the full-sample estimate and the estimate obtained after data splitting and imputation. Then, under the following conditions:

**Condition 1.** *There exists a positive constant $\alpha > 0$ such that for any $M \in \mathcal{M}$ and any $\varepsilon > 0$, there exist a constant $\delta_0$ and an integer $K_0$ such that*

$$P\left( \left| \frac{Q_K(M)}{K^{-\alpha}} \right| \geq \delta_0 \right) \leq \varepsilon \quad \forall K \geq K_0,$$

we can show that

**Theorem 1.** *As $n \to \infty$ and $K \to \infty$,*

$$V_K(M) - L(M) - \Lambda = O_p \left( \frac{1}{n} \vee \frac{1}{K^{(1+\alpha)/2}} \vee \frac{1}{K^\alpha} \right) \tag{8}$$

*uniformly, where $\Lambda = \frac{2}{n(n-1)} \sum_{i<j} p_{ij}(1 - p_{ij})$, and $\vee$ denotes the operator that returns the largest of two values.*

Theorem 1 suggests that the validation score $V_K(M)$ is a consistent estimator of $L(M) + \Lambda$ with an error rate of $\frac{1}{n} \vee \frac{1}{K^{(1+\alpha)/2}} \vee \frac{1}{K^\alpha}$. Note that $\Lambda$ does not depend on $M$, so the validation score function $V_K(\cdot)$ and the loss function $L(\cdot)$ are asymptotically parallel in probability. Thus, the probability that the minimizer of $V_K(M)$ approximately minimizes $L(M)$ is high within a neighborhood of $M_0$. Theorem 1 essentially establishes the consistency of the CV-imputation score, ensuring that the score for each candidate model is close to its true loss up to a constant. The proof of Theorem 1 is provided in the appendix.

It is worth noting that Condition 1 specifies that the maximum $K$-fold optimism bias is bounded by a polynomial rate of $K^{-\alpha}$. The polynomial order $\alpha$ is determined by the complexity of the underlying graphon model and the efficiency of the estimation methods. For instance, in the case of an Erdős–Rényi model (Erdos and Renyi, 1959) where $f(\mu_i, \mu_j) \equiv p$, we have $\alpha = 1$ when using the simple averaging estimator and letting $K \asymp n$. Unlike many assumptions that are not verifiable, $Q_K(M)$ can be verified computationally, as both $\hat{\mathbf{P}}(M|\mathbf{A})$ and $\hat{\mathbf{P}}_k(M)$ are accessible from the data. In Figure S.3 in the Appendix, the validation of Condition 1 is provided. Further discussion of Condition 1 can be found in Section S.10.

## 5 EMPIRICAL STUDIES

To compare the empirical performance of our method to the edge cross-validation method proposed in Li et al. (2020a), we investigate their performance using data generated from four graphon models introduced in (Chan and Airoldi, 2014). Figure 2 illustrates the probability matrices $\mathbf{P}$ of 200 nodes generated for Graphons 1 to 4. Graphons 1 and 2 generate dense networks, while Graphons 3 and 4 yield sparse networks. Graphons 1, 3, and 4 produce low-rank probability matrices, while Graphon 2 produces full-rank probability matrices. All results are averaged over 100 replications. The experiments are conducted on a machine equipped with a 40-core CPU and 192 GB of RAM. The implemtation issues or our methods are detailed in Section S.4 in the Appendix.

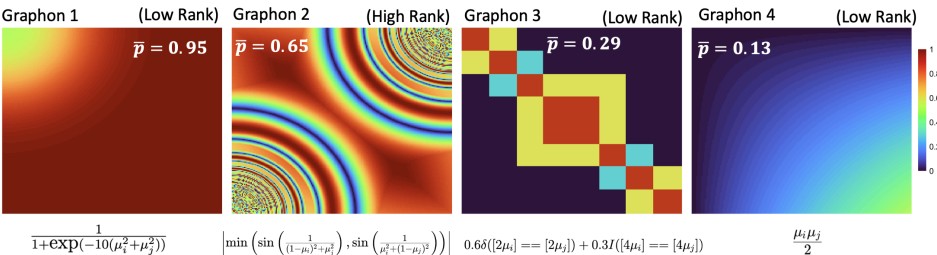

Figure 2: The heatmaps of $\mathbf{P}$ generated by Graphons 1 to 4 are displayed from left to right.

To evaluate the empirical performance of our method, we use it to select the tuning parameter for four state-of-the-art graphon estimation methods: (1) neighborhood smoothing (NS) method (Zhang et al., 2017), (2) sort-and-smooth (SAS) method (Chan and Airoldi, 2014), (3) universal singular value thresholding (USVT) method (Chatterjee, 2015), and (4) iterative connecting probability estimation method (ICE) (Qin et al., 2021). All four methods make different conditions about the underlying model. In each of these estimation methods, the parameter $M$, although with different meanings, governs the trade-off between goodness-of-fit and model complexity. For instance, in the NS method, $M$ represents the neighborhood size, while in the SAS method, it denotes the number of blocks. We apply our method with the selected tuning parameter to the synthetic datasets generated from the four aforementioned graphon functions. This enables us to compare the estimated graphons with the true underlying graphon functions and assess the performance of our method in recovering the underlying structure of the graphons using the mean squared error (MSE) measurement denoted by $L(M)$.

In Table 1, we present the estimation accuracy measured by $L(M)$ for $n = 200$. The table illustrates that for all five estimation methods, our method and ECV select $M$ resulting in lower MSE values compared to the default selection. This underscores the significance of tuning $M$ for all five estimation methods. Moreover, CV-imputation method consistently selects models with smaller MSE values compared to those chosen by ECV for all five methods and all synthetic datasets.

Table 1: The mean $\pm$ standard deviation of MSE across 100 replicates are calculated using $M$ selected by CV-imputation, ECV, and default selection. To facilitate comparison, all values are multiplied by 100. Note that ICE does not have a default model setup, so the default ICE results are not shown in this table.

| Graphon ID | Graphon 1 | Graphon 2 | Graphon 3 | Graphon 4 |
|---|---|---|---|---|
| CV-imputation (NS) | **0.51 ± 0.07** | **2.13 ± 0.15** | **0.79 ± 0.07** | **1.05 ± 0.06** |
| ECV (NS) | 9.15 ± 19.25 | 3.82 ± 0.21 | 3.07 ± 0.95 | 1.06 ± 0.10 |
| Default NS (M=1) | 39.05 ± 3.33 | 2.75 ± 0.16 | **0.74 ± 0.04** | 1.06 ± 0.10 |
| CV-imputation (USVT) | **0.28 ± 0.03** | **2.99 ± 0.19** | **0.61 ± 0.10** | **0.75 ± 0.05** |
| ECV (USVT) | 0.60 ± 0.09 | 5.06 ± 0.27 | 1.18 ± 0.02 | 1.08 ± 0.74 |
| Default USVT (M=0.01) | 0.60 ± 0.09 | 5.06 ± 0.25 | 1.18 ± 0.02 | 2.79 ± 0.26 |
| CV-imputation (SAS) | **1.69 ± 0.11** | **8.43 ± 0.22** | **12.77 ± 0.20** | **1.49 ± 0.10** |
| ECV (SAS) | 1.72 ± 0.12 | 8.47 ± 0.27 | 12.86 ± 0.23 | 1.53 ± 0.11 |
| Default SAS (M=[n/logn]) | 1.93 ± 0.15 | 8.77 ± 0.23 | 13.64 ± 0.60 | 1.89 ± 0.14 |
| CV-imputation (ICE) | **0.31 ± 0.03** | **2.69 ± 0.24** | **0.50 ± 0.06** | **0.82 ± 0.06** |
| ECV (ICE) | 0.32 ± 0.05 | 3.05 ± 0.55 | 0.53 ± 0.06 | 0.86 ± 0.06 |

Figure 3 provides a comparison of the end-to-end computational cost between our method (CV-imputation) and ECV. Here, the reported CPU time includes both the cost of fitting the graphon estimators (NS, USVT, SAS, ICE) and the cross-validation overhead (fold construction, masking, prediction on validation edges, and loss aggregation). It is clear that our method consistently outperforms ECV in terms of speed across all tested configurations. To isolate estimator fitting overhead from the cross-validation mechanism itself, Figure S.7 further reports the cross-validation–only runtime (with the graphon estimation time subtracted out). The cross-validation–only results show that CV-imputation also remains substantially faster than ECV at this level, confirming that the observed speedup is indeed driven by our CV-imputation scheme.

To evaluate the consistency of our model selection method as network size increases, we compare mean square validation error $V_K(\cdot)$ and mean squared error (MSE) $L(\cdot)$. Both errors are standardized using the function $\frac{L - \min(L)}{\max(L) - \min(L)}$ for better visualization. In Figure 4, we focus on the performance of CV-imputation for the NS estimator. Additional plots for the other three estimation methods are provided in Figure S.4, Figure S.5, and Figure S.6 in the supplementary materials. We examine the influence of parameter $M$ ranging from 0.5 to 5 with an increment of 0.5. The number of nodes in the network ranges from 50 to 200 in steps of 50 to allow for an evaluation of performance as the network size increases towards asymptotic levels.

Figure 4 demonstrates that the model selected by our method rapidly converges to the optimal model $M_o$. When $n = 50$, the model selected by our method aligns with the optimal model selected by MSE for Graphons 1 and 2, a pattern that holds across all graphons when $n = 200$. Similar trends were observed for all the other methods. These simulations demonstrate that CV-imputation maintains rank consistency in model selection for any given estimation approach.

Similar to traditional cross-validation (CV) techniques, CV-imputation offers a model-agnostic measure for assessing a model's generalization to new data. The term "agnostic" in this context indicates that our method is unbiased towards any specific model or estimation technique. It functions independently of particular conditions or methods and can be universally applied, irrespective of the underlying model structure or estimation process.

The upper panel of Figure 5 demonstrates the consistent superiority of our method over ECV across various graphon configurations, with more pronounced benefits observed for smaller network sizes ($n$). It is important to note that Figure 5 evaluates the method selection task, choosing which estimation method (NS, USVT, SAS, or ICE) with its optimally tuned hyperparameter should be used, rather than hyperparameter tuning for a single method. Notably, at $n = 200$, our method achieves a 100% accuracy rate in selecting the best candidate model – that is, the one with the lowest estimation Mean Squared Error (MSE) among all estimators obtained by the five given estimation methods. Here,

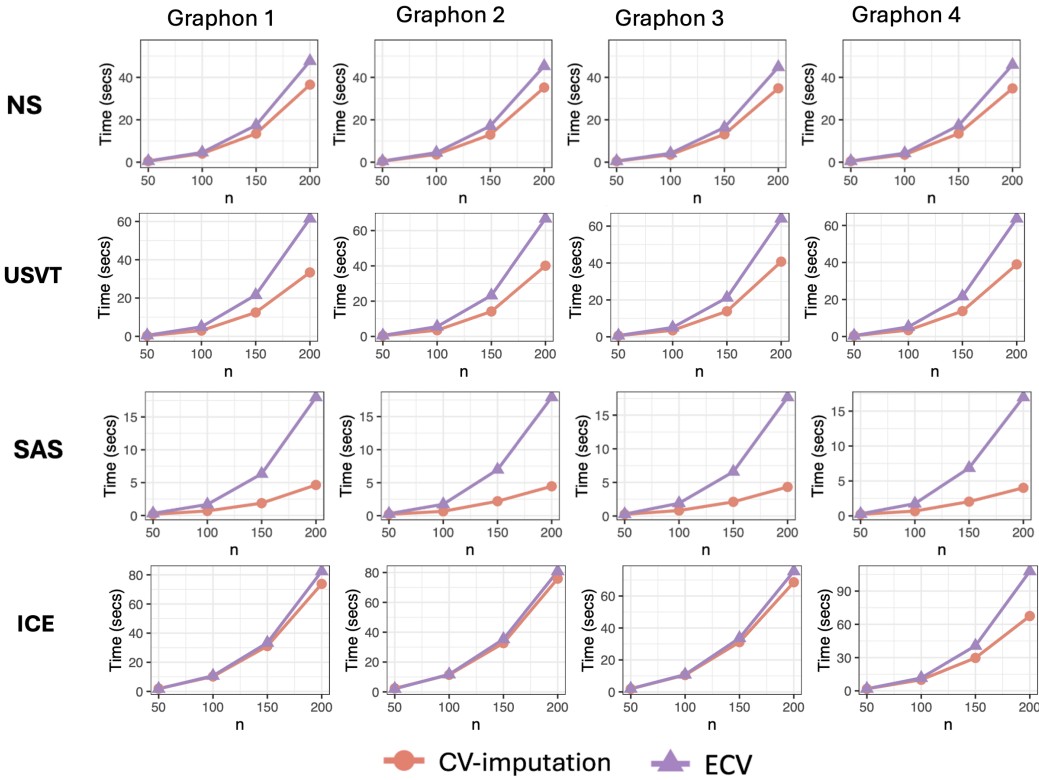

Figure 3: The plots display the average computational time (in seconds) for Graphons 1 to 4 with $n \in \{50, 100, 150, 200\}$, arranged from left to right. The panel from top to bottom corresponds to the NS method, the USVT method, the SAS method, and the ICE method, respectively.

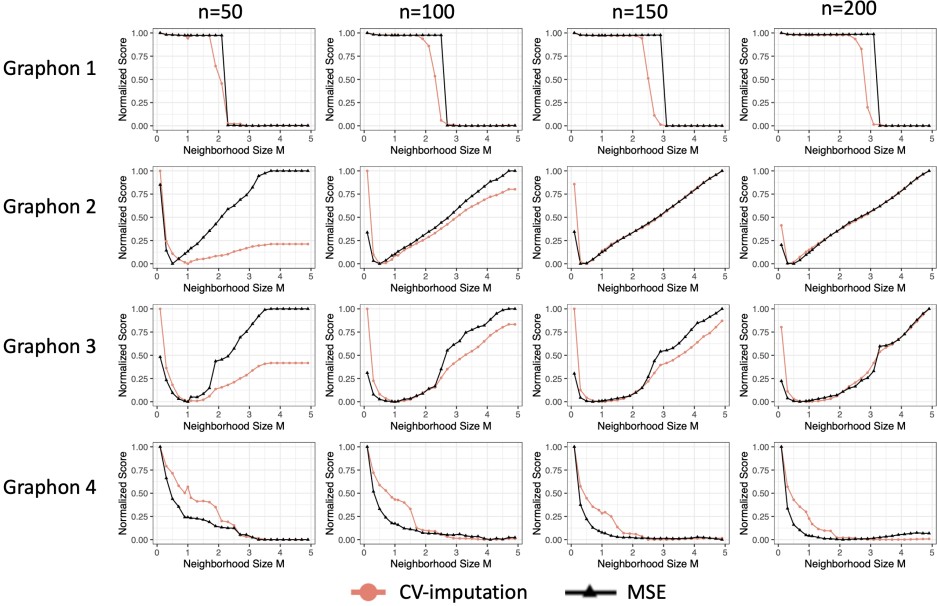

Figure 4: Plotted here are the scores for CV-imputation (red) and MSE (black) under varying values of the tuning parameters for the NS method. We vary the neighborhood size parameter $M$ from 0.5 to 5 with increments of 0.5, while the number of nodes $n$ ranges from 50 to 200 with an increment of 50. Each row corresponds to a specific graphon function listed in Figure 2.

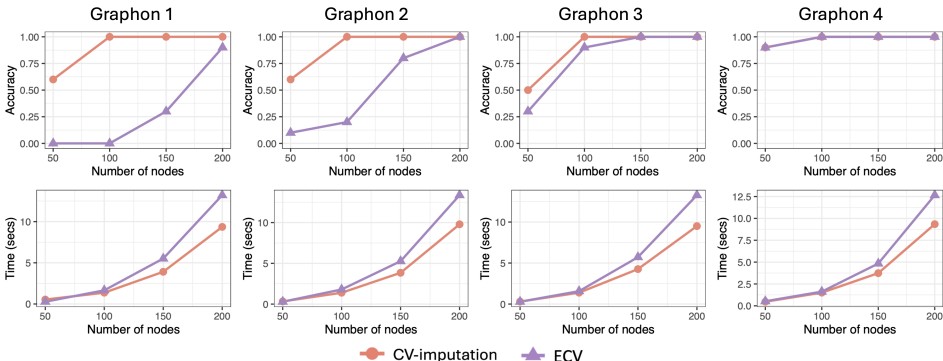

Figure 5: Method selection performance across different graphon designs. The plots display the average selection accuracy and computational time (in seconds) for Graphons 1 to 4, arranged from left to right. The top panel illustrates CV-imputation's model selection accuracy as $n$ increases from 50 to 200 in steps of 50, while the bottom panel shows the corresponding computational time. Here, accuracy is defined as the percentage of cases where the model with the smallest mean squared error (MSE) is selected from the top-tuned estimators obtained using NS, SAS, USVT, and ICE, respectively.

"accuracy" denotes the percentage of cases where our method successfully identifies the optimal candidate model. This indicates that our method is particularly efficient for model selection compared to ECV. Furthermore, the lower panel of Figure 5 displays the computational efficiency of our method in relation to ECV. The reported time measures only the computational cost of the method selection step itself, excluding the time spent on hyperparameter tuning for each individual method. Our approach consistently outperforms ECV in terms of speed across all configurations. These results collectively highlight the effectiveness and efficiency of our approach for model selection.

# 6 CASE STUDY: MODEL SELECTION IN LINK PREDICTION

We applied our method to four networks: the drug-disease co-occurrence network with 280 nodes and 952 edges; the political blogs network, consisting of 1,222 nodes and 16,714 edges (Adamic and Glance, 2005); the coauthorship network for scientists in network science, which contains 1,589 nodes and 2,742 edges (Newman, 2006); and the yeast protein-protein interaction network, comprising 2,617 nodes and 11,855 edges (Von Mering et al., 2002). These example networks vary in size, including moderate-sized networks suitable for detailed analysis and larger networks that illustrate real-world computational challenges. Analyzing these networks allows us to evaluate the capability of CV-imputation to accurately identify existing connections and potentially uncover new insights.

For all networks, we used both CV-imputation and ECV to first tune each estimation method to obtain the optimal $M_0^{\text{method}}$, and then compared across the aforementioned candidate estimation methods—NS, ICE, SAS, and USVT—to identify the best graphon estimate $M_0$. Since NS and ICE are highly computationally intensive, we did not include them as candidate methods for large networks with over 1,000 nodes.

## 6.1 MODERATE SIZE NETWORK: TEXT CO-OCCURRENCE NETWORK FOR COVID-19

In this study, the abstracts of 5,496 publications were retrieved from the Medline database using "COVID-19" as the query term and a time window of January 1, 2020, to April 30, 2020. After conducting data cleaning and entity annotation procedures, we identified entities related to drugs or diseases and constructed a co-occurrence network consisting of 280 nodes and 952 edges [1]. The network is sparse, with a density of only 0.02. The co-occurrence network, depicted in Figure 6(a), highlights two central nodes: COVID-19 and hydroxychloroquine. In what follows, we show our estimation results using neighborhood smoothing method (Zhang et al., 2017) for variable neighbourhood size.

---

[1]The generated data can be freely downloaded from the following website: https://drive.google.com/file/d/1yvtE58n4Pz6nQT7AWtdzQEHb39Bxy0MG/view?usp=sharing.

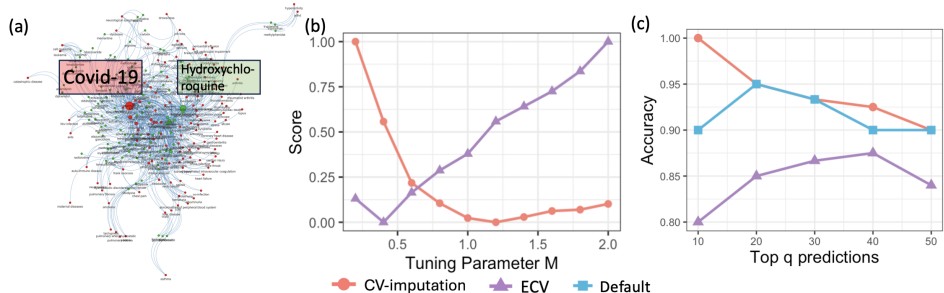

Figure 6: (a) Disease-drug co-occurrence network with red and green nodes representing diseases and drugs, respectively. (b) CV-imputation score and ECV score with different models $M$. (c) Accuracy of CV-imputation and ECV for different $q$.

Figure 6(b) illustrates the relationship between the average prediction error of K-fold cross-validation (CV) and the neighborhood size parameter $M$, ranging from 0.2 to 2. The minimum average prediction error $L(M)$ occurs at $M = 1.2$, contrasting with ECV, which achieves its minimum at $M = 0.4$.

To evaluate the empirical performance of the CV-imputation method against ECV, we utilize testing data extracted from articles published between May 1, 2020, and May 15, 2020. A predicted link is deemed correct if it also exists in the testing network. Figure 6(c) illustrates the proportion of the correctly predicted links among the top $q$ predicted links with $q$ ranging from 10 to 50, for $M = 1.2$ (selected by CV-imputation) and $M = 0.4$ (selected by ECV) respectively. Generally, as more predicted links are included, the accuracy tends to decrease. However, our method consistently outperforms ECV in terms of prediction accuracy. Furthermore, our method boasts a computational cost of 56.76 seconds, which is more efficient than the 71.82 seconds required for ECV, as it eliminates the need for the time-consuming matrix completion step. Remarkably, among all current unlinked node pairs, the third highest predicted probability corresponds to a link between COVID-19 and ledipasvir, a drug primarily developed for treating hepatitis C virus infections. Our findings suggest a potential repurposing of ledipasvir to treat COVID-19. Recent research findings support our discovery, indicating that ledipasvir, when combined with sofosbuvir, exhibits significant potential to inhibit SARS-CoV-2 replication (Pirzada et al., 2021). This medical condition was further confirmed by a recently completed phase-3 clinical trial. This particular application underscores the importance of our approach in drug repurposing efforts.

## 6.2 LARGE SIZE NETWORKS: POLITICAL BLOGS NETWORK, COAUTHORSHIP NETWORK, AND PROTEIN INTERACTION NETWORK.

To evaluate the prediction accuracy, we randomly sampled 10% of the node pairs from each network, with the connectivity of the sampled node pairs used as testing data to assess link prediction accuracy and computational efficiency. Three large networks: the political blogs network (PolBlog), the coauthorship network for scientists in network science (NetSci), and the yeast protein-protein interaction network (Yeast) were compared on the area under the curve (AUC) metric, and the average time taken in minutes metric.

As shown in Table 2, CV-imputation significantly outperforms the ECV method for the PolBlog and NetSci networks while demonstrating comparable prediction accuracy for the Yeast network. Moreover, since CV-imputation eliminates the highly computationally intensive matrix completion step in each fold, it substantially reduces computational costs, making it significantly more efficient.

Table 2: AUC (average ± standard deviation) and computational time in minutes (average ± standard deviation) of CV-imputation and ECV, over 100 replications.

|  |  | PolBlog | NetSci | Yeast |
|---|---|---|---|---|
| AUC | CV-imputation | **0.88 ± 0.01** | **0.72 ± 0.01** | 0.80 ± 0.02 |
|  | ECV | 0.80 ± 0.02 | 0.70 ± 0.01 | 0.80 ± 0.02 |
| Time (seconds) | CV-imputation | **56.90 ± 0.13** | **51.01 ± 3.96** | **240.90 ± 16.22** |
|  | ECV | 258.65 ± 2.11 | 771.23 ± 10.00 | 6021.12 ± 18.72 |

## 7 Conclusions and Discussions

Our proposed method, denoted as CV-imputation, offers several key advantages that make it a valuable tool for analyzing network data. Firstly, CV-imputation is versatile as it does not assume any specific form for the graphon function, making it broadly applicable across different types of networks. This flexibility enables researchers to apply CV-imputation to various network structures without conforming to a specific model.

Moreover, CV-imputation is supported by rigorous theoretical foundations, showing that the minimizer of CV-imputation asymptotically converges to the minimizer of the mean squared error (MSE) in our model. This theoretical basis enhances confidence in the accuracy and consistency of CV-imputation results.

In terms of computation efficiency, CV-imputation outperforms existing methods by eliminating costly singular value decomposition (SVD) steps, making it particularly suitable for analyzing large networks. Our simulation studies have confirmed the superior performance of CV-imputation, further highlighting its effectiveness in practical applications.

Beyond graphons, CV-imputation may extend to broader network models that preserve the required edge-independence structure, such as latent-space networks and generalized sparse graphons. Section S.9 provides preliminary empirical support and discusses how our theoretical results may extend to these settings. Establishing formal guarantees in these settings remains an important direction for future work. However, our method can not be extended to models with temporal or sequential dependence since they violate edge independence.

In summary, CV-imputation stands out as an efficient, versatile, and theoretically grounded method for analyzing networks. Its user-friendly implementation and lack of tuning requirements make it a practical choice for various network analysis tasks.

## Use of LLMs

During the preparation of this manuscript, the authors used ChatGPT (https://openai.com/) solely for improving the language, readability, and correcting the grammar.

## Acknowledgments

This work was partially supported by the U.S. National Science Foundation [DMS-1925066, DMS-1903226, DMS-2124493, DMS-2311297, DMS-2319279, DMS-2318809] and the National Institutes of Health [NIH R01GM152814]. The authors thank reviewers for helpful discussions and support.

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
