# Appendix

## S.1   PROOF

### S.1.1   PROOF OF LEMMA 1

*Proof.* Here, we derive the conditional distribution of $\mathbf{A}_{ij}^{[-k]}$ given $\mathbf{P}$. To ease the mathematical representation, we omit this condition in the probability function. Recall that

$$
\begin{aligned}
P((v_i, v_j) \in \mathcal{S}_k) &= w_k \\
P(b_{ij} = 1) &= \theta.
\end{aligned}
$$

Hence,

$$
\begin{aligned}
P(\mathbf{A}_{ij}^{[-k]} = 1) &= P((v_i, v_j) \notin \mathcal{S}_k)P(a_{ij} = 1) + P((v_i, v_j) \in \mathcal{S}_k)P(b_{ij} = 1) \\
&= w_k \theta + (1 - w_k)p_{ij}
\end{aligned}
$$

We have

$$
\begin{aligned}
&P(\mathbf{A}_{ij}^{[-k]} = 1, a_{i'j'} = 1) \\
&= P((v_i, v_j) \in \mathcal{S}_k)P(b_{ij} = 1, a_{i'j'} = 1) + P((v_i, v_j) \notin \mathcal{S}_k)P(a_{ij} = 1, a_{i'j'} = 1) \\
&= P((v_i, v_j) \in \mathcal{S}_k)P(b_{ij} = 1)P(a_{i'j'} = 1) + P((v_i, v_j) \notin \mathcal{S}_k)P(a_{ij} = 1)P(a_{i'j'} = 1) \\
&= P(a_{i'j'} = 1)(w_k \theta + (1 - w_k)p_{ij}) \\
&= P(\mathbf{A}_{ij}^{[-k]} = 1)P(a_{i'j'} = 1)
\end{aligned}
$$

Therefore, the connectivity probabilities of the node pairs in a training graph are mutually independent of the connectivity probabilities for node pairs in $\mathcal{S}_k$.

$\square$

### S.1.2   PROOF OF THEOREM 1

Since the left-hand side of equation (8) is given by $|V_K(M) - L_K(M) - \Lambda + L_K(M) - L(M)|$, where

$$
L_K(M) = \frac{2}{n(n-1)} \sum_{k=1}^{K} \sum_{(v_i, v_j) \in \mathcal{S}_k} (\hat{p}_{ij}^{[k]}(M) - p_{ij})^2,
$$

to prove Theorem 1, we only need to prove the following parts:

$$
\Delta_1 \overset{\triangle}{=} V_K(M) - L_K(M) - \Lambda = O_p\left(\frac{1}{n} \vee \frac{1}{K^{(1+\alpha)/2}}\right) \quad \text{and} \tag{9}
$$

$$
\Delta_2 \overset{\triangle}{=} L_K(M) - L(M) = O_p\left(\frac{1}{K^\alpha}\right). \tag{10}
$$

Equation (9) follows with the assistance of Lemma S.1 and Lemma S.2, while (10) is obtained from Lemma S.3.

**Lemma S.1.** *Under the condition of Lemma 1, conditional on the probability matrix* $\mathbf{P}$, $\Lambda$ *is an unbiased estimator for* $V_K(M) - L_K(M)$.

*Proof.*

$$V_K(M) - L_K(M) - \Lambda =$$

$$\frac{2}{n(n-1)} \sum_{k=1}^{K} \sum_{(v_i,v_j) \in \mathcal{S}_k} \left( (p_{ij} - a_{ij})^2 + 2(\hat{p}_{ij}^{[k]}(M) - p_{ij})(p_{ij} - a_{ij}) \right)$$

Lemma 1 implies that for any $(i,j)$ where $(v_i, v_j) \in \mathcal{S}_k$, $a_{ij}$ is independent of $\hat{p}_{ij}^{[k]}(M)$. Thus,

$$\mathbb{E}(V_K(M) - L_K(M)) = \frac{2}{n(n-1)} \sum_{k=1}^{K} \sum_{(v_i,v_j) \in \mathcal{S}_k} p_{ij}(1 - p_{ij}) = \Lambda,$$

where the expectation is conditional expecation with any given $\mathbf{P}$. $\qquad\square$

**Lemma S.2.** *Under the condition of Lemma 1 and Condition 1,* $\mathrm{Var}(V_K(M) - L_K(M)) = O(1/n^2 \vee 1/K^{1+\alpha})$

*Proof.* By the law of total variance:

$$\mathrm{Var}(V_K(M) - L_K(M)) = \mathbb{E}[\mathrm{Var}(V_K(M) - L_K(M) \mid \{\mathcal{S}_k\})] \\ + \mathrm{Var}(\mathbb{E}[V_K(M) - L_K(M) \mid \{\mathcal{S}_k\}]).$$

From Lemma S.1, $\mathbb{E}[V_K(M) - L_K(M) \mid \{\mathcal{S}_k\}] = \Lambda$, which does not depend on the partition $\{\mathcal{S}_k\}$. Therefore, we only analyze the first term.

Conditioned on the partition $\{\mathcal{S}_k\}$, simple algebra yields $\mathrm{Var}(V_K(M) - L_K(M) \mid \{\mathcal{S}_k\}) = (\frac{2}{n(n-1)})^2 (\Delta_3 + \Delta_4)$, where

$$\Delta_3 = \sum_{k=1}^{K} \sum_{(v_i,v_j) \in \mathcal{S}_k} \mathrm{Var}\left( (p_{ij} - a_{ij})(2\hat{p}_{ij}^{[k]}(M) - p_{ij} - a_{ij}) \right) \quad and$$

$$\Delta_4 = \frac{1}{K-1} \sum_{k_1,k_2=1}^{K} \sum_{(v_i,v_j) \in \mathcal{S}_{k_1} \neq (v_q,v_r) \in \mathcal{S}_{k_2}} \\ \mathrm{Cov}\left( (p_{ij} - a_{ij})(2\hat{p}_{ij}^{[k_1]}(M) - p_{ij} - a_{ij}), (p_{qr} - a_{qr})(2\hat{p}_{qr}^{[k_2]}(M) - p_{qr} - a_{qr}) \right).$$

Notice that

$$\mathbb{E}[(p_{ij} - a_{ij})(2\hat{p}_{ij}^{[k]}(M) - p_{ij} - a_{ij})] = p_{ij} - p_{ij}^2$$

A straightforward simplification gives

$$\Delta_3 = \sum_{k=1}^{K} \sum_{(v_i,v_j) \in \mathcal{S}_k} p_{ij}(1 - p_{ij}) \mathbb{E}(1 - 2\hat{p}_{ij}^{[k]}(M))^2,$$

which is of order $O(n^2)$, since $p_{ij}(1 - p_{ij})\mathbb{E}(1 - 2\hat{p}_{ij}^{[k]}(M))^2$ is bounded by a constant.

Meanwhile, $\Delta_4 = \frac{1}{K-1} \sum_{k_1,k_2=1}^{K} \sum_{(v_i,v_j) \in \mathcal{S}_{k_1} \neq (v_q,v_r) \in \mathcal{S}_{k_2}} (\Delta_5 + \Delta_6)$, where

$$\Delta_5 = \mathbb{E}[(p_{ij} - a_{ij})(p_{qr} - a_{qr})(2\hat{p}_{ij}^{[k_1]}(M) - 1)(2\hat{p}_{qr}^{[k_1]}(M) - 1)] + \\ \mathbb{E}[(p_{ij} - a_{ij})(p_{qr} - a_{qr})(2\hat{p}_{ij}^{[k_2]}(M) - 1)(2\hat{p}_{qr}^{[k_2]}(M) - 2\hat{p}_{qr}^{[k_1]}(M)] \quad and$$

$$\Delta_6 = \mathbb{E}[(p_{ij} - a_{ij})(p_{qr} - a_{qr})(2\hat{p}_{ij}^{[k_1]}(M) - 2\hat{p}_{ij}^{[k_2]}(M))(2\hat{p}_{qr}^{[k_2]}(M) - 2\hat{p}_{qr}^{[k_1]}(M))]$$

Lemma 1 implies that $a_{ij}$ is independent of $a_{qr}$, $\hat{p}_{ij}^{[k_1]}(M)$, and $\hat{p}_{qr}^{[k_1]}(M)$, $a_{qr}$ is independent of $\hat{p}_{ij}^{[k_2]}(M)$, and $\hat{p}_{qr}^{[k_2]}(M)$. A straightforward mathematical derivation then yields $\Delta_5 = 0$.

Now, let us consider different setups for $\Delta_6$. If $k_1 = k_2$, which implies $\hat{p}_{qr}^{[k_2]}(M) = \hat{p}_{qr}^{[k_1]}(M)$, then we have $\Delta_6 = 0$. Otherwise, we apply the Cauchy-Schwarz inequality step-by-step and get

$$
\begin{aligned}
|\Delta_6| &\leq \mathbb{E}\left|(p_{ij} - a_{ij})(p_{qr} - a_{qr})(2\hat{p}_{ij}^{[k_1]}(M) - 1)\right| \cdot \\
&\quad \mathbb{E}\left|(2\hat{p}_{ij}^{[k_2]}(M) - 2\hat{p}_{ij}^{[k_1]}(M))(2\hat{p}_{qr}^{[k_2]}(M) - 2\hat{p}_{qr}^{[k_1]}(M))\right| \\
&\leq 24\left(\mathbb{E}\left[(\hat{p}_{ij}^{[k_2]}(M) - \hat{p}_{ij}^{[k_1]}(M))\right]^2 + \mathbb{E}\left[(\hat{p}_{qr}^{[k_2]}(M) - \hat{p}_{qr}^{[k_1]}(M))\right]^2\right)
\end{aligned}
\tag{11}
$$

Consequently, we have

$$
\begin{aligned}
\mathbb{E}[|\Delta_4| \big| \{\mathcal{S}_k\}] &= \mathbb{E}\left[\left|\frac{1}{K-1}\sum_{k_1 \neq k_2}\sum_{(v_q,v_r)\in\mathcal{S}_{k_1}\neq(v_i,v_j)\in\mathcal{S}_{k_2}}\Delta_6\right|\Big|\{\mathcal{S}_k\}\right] \\
&\leq \frac{1}{K-1}\mathbb{E}\left[\sum_{k_1 \neq k_2}\sum_{(v_q,v_r)\in\mathcal{S}_{k_1}\neq(v_i,v_j)\in\mathcal{S}_{k_2}}|\Delta_6|\big|\{\mathcal{S}_k\}\right] \\
&\leq \frac{24n^2}{K-1}\mathbb{E}[\sum_{k_1}\sum_{(v_q,v_r)\in\mathcal{S}_{k_1}}Q_K(M)] \\
&\leq 24\frac{n^4}{K-1}\mathbb{E}[Q_K(M)],
\end{aligned}
$$

where the expectation in $\mathbb{E}[Q_K(M)]$ is taken over (1) $\mathbf{A}^{[-k]}$ where the randomness comes from Bernoulli realizations and latent position $\{\mu_i\}$, and (2) $\{\mathcal{S}_k\}$.

Thus, $\Delta_4$ converges in an order of $O\left(\frac{n^4}{K^{1+\alpha}}\right)$ under Condition 1. Hence,

$$
\text{Var}(V_K(M) - L_K(M)) = \left(\frac{2}{n(n-1)}\right)^2(\Delta_3 + \Delta_4),
$$

converges in an order of $O\left(1/n^2 \vee 1/K^{1+\alpha}\right)$. $\qquad\square$

Lemmas S.1 and S.2 ensure that equation (9) holds by Chebyshev's inequality. Next, we show in Lemma S.3 that equation (10) holds.

**Lemma S.3.** $L_K(M) - L(M) = O_p(\frac{1}{K^\alpha})$ if Condition 1 holds.

*Proof.* Based on the triangle inequality, we can show that

$$
\begin{aligned}
\mathbb{E}|L_K(M) - L(M)| &= \mathbb{E}\left|\frac{2}{n(n-1)}\sum_{k=1}^{K}\sum_{(v_i,v_j)\in\mathcal{S}_k}[(\hat{p}_{ij}^{[k]}(M) - p_{ij})^2 - (\hat{p}_{ij}(M) - p_{ij})^2]\right| \\
&\leq \frac{2}{n(n-1)}\sum_{k=1}^{K}\sum_{(v_i,v_j)\in\mathcal{S}_k}\mathbb{E}\left|(\hat{p}_{ij}^{[k]}(M) - p_{ij})^2 - (\hat{p}_{ij}(M) - p_{ij})^2\right| \\
&\leq \frac{2}{n(n-1)}\sum_{k=1}^{K}\sum_{(v_i,v_j)\in\mathcal{S}_k}\mathbb{E}(\hat{p}_{ij}^{[k]}(M) - \hat{p}_{ij}(M))^2 \\
&\leq \mathbb{E}Q_K(M),
\end{aligned}
\tag{12}
$$

Hence, under Condition 1, we can show that $L_K(M) - L(M) = O_p(\frac{1}{K^\alpha})$ $\qquad\square$

## S.2 Detailed Algorithm

## S.3 Detailed Algorithm

---

**Algorithm 1** K-fold Graphon Cross-Validation with Random Imputation

---

**Input:** (1) adjacency matrix of a single observed graph $\mathbf{A}$;

   (2) candidate models $\mathcal{M}$;

   (3) prespecified parameters $\theta$.

**Partition:** Randomly partition $\{(v_i, v_j) \in \mathcal{V} \times \mathcal{V} : i < j\}$ into $K$ subsets of approximately equal size. Each subset is referred to as a fold.

**for** each $M \in \mathcal{M}$ **do**

   **Iterate over folds:** for $k = 1$ to $K$:

      **Imputation:** Construct the training adjacency matrix $\mathbf{A}^{[-k]}$ based on (4).

      **Training:** Train $M$ using $\mathbf{A}^{[-k]}$ to compute $\hat{\mathbf{P}}(M \mid \mathbf{A}^{[-k]})$.

      **Debias:** Obtain debiased estimator $\hat{\mathbf{P}}_k(M) = \frac{\hat{\mathbf{P}}(M|\mathbf{A}^{[-k]}) - w_k \theta}{1 - w_k}$.

**end for**

**Validation score:** Calculate the validation score on the validation set for $M$ using $V_K(M) = \frac{2}{n(n-1)} \sum_{k=1}^{K} \sum_{(v_i, v_j) \in \mathcal{S}_k} (\hat{p}_{ij}^{[k]}(M) - a_{ij})^2$.

**Output:** Selected optimal model $M_V$ which minimizes $V_K(M)$.

---

## S.4 Implementation issues

The number of folds $K$ in CV-imputation may influence the accurate assessment of a model's performance. Opting for a higher number of folds, such as leave-one-out CV, can enhance accuracy by utilizing more training data but may escalate computational costs, particularly with larger datasets. Conversely, employing fewer folds reduces the computational burden but may lead to less precise performance estimates due to limited data usage during model training. After conducting thorough research and numerical robustness experiments, we determined that the accuracy of CV-imputation is not substantially affected by the specific value of $K$, see Figure S.1. Therefore, for the sake of simplicity and to ensure asymptotic consistency, we set $K = n/5$ in this paper.

In comparison to tuning the number of folds $K$ in CV-imputation, the random imputation parameter $\theta$ holds greater importance as it significantly influences the imputation process and the quality of the training network. The choice of $\theta$ dictates how validation entries are filled in, resulting in different patterns and characteristics in the training network. By adjusting $\theta$, researchers can customize the imputation process to potentially enhance the representativeness of imputed values and maintain the dataset's integrity. Therefore, tuning $\theta$ plays a crucial role in optimizing the CV-imputation process, improving the quality of the training network, and ensuring that the estimated out-of-sample error aligns with that of the original network, $\frac{1}{n} \sum_{i<j} (\theta - p_{ij})^2$, which is minimized at $\theta = \frac{2}{n(n-1)} \sum_{(v_i, v_j) \in \mathcal{S}_k : i<j} p_{ij}$. The minimizer can be estimated by the network density: $\bar{a} = \frac{\sum_{i<j} a_{ij}}{n(n-1)/2}$. When calculating the network density, we leave out the information contained in the validation data. We thus set $\theta = \frac{\sum_{v_i v_j \in \mathcal{S}_{-k}} a_{ij}}{|\mathcal{S}_{-k}|}$. In addition, our empirical results in Figure S.2 show that the performance of CV-imputation is always competitive when $\theta$ is selected to be the network density, which empirically confirms our proposal.

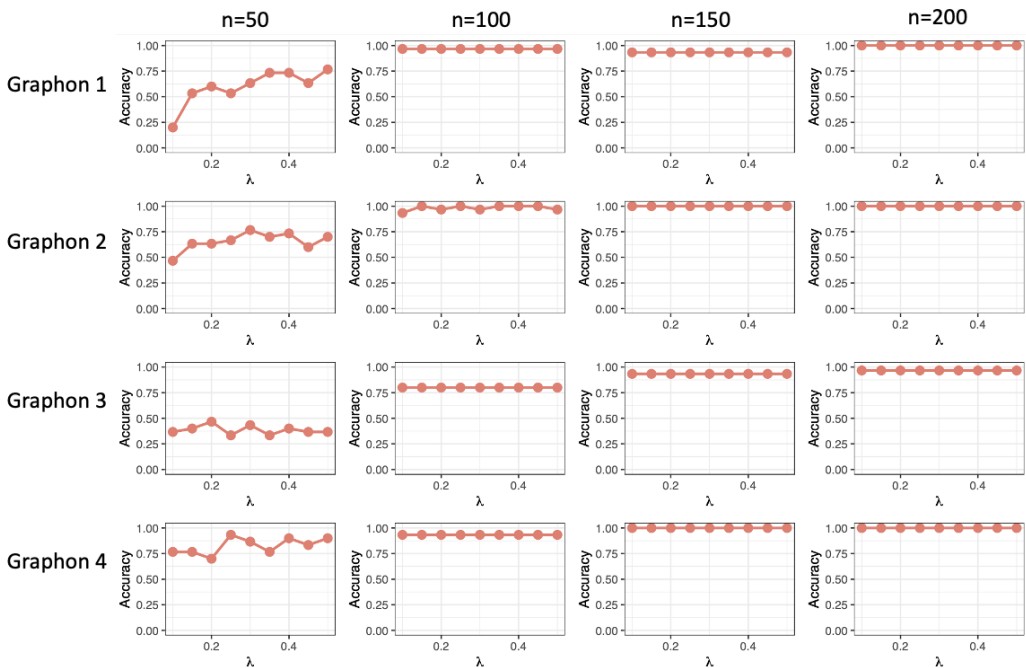

Figure S.1: The plots illustrate selection accuracy—defined as the percentage of instances where our method correctly identifies the optimal candidate model—against $\lambda = K/n$ for sample sizes $50, 100, 150$, and $200$ (arranged from left to right) and Graphon models 1 to 4 (organized from top to bottom). The impact of $\lambda$ is nearly negligible across all simulated examples.

## S.5 ROBUST ANALYSIS FOR $K$ AND $\theta$

## S.6 VALIDATION OF CONDITION 1

Figure S.3 illustrates the behavior of $\log Q_K$ as $\log K$ increases across various graphon settings and estimation methods. The linear relationship between $\log(Q_K(M))$ and $\log K$ indicates that $Q_K(M)$ adheres to a polynomial bound as $K$ grows.

## S.7 ADDITIONAL HYPERPARAMETER SELECTION RESULTS

## S.8 COMPUTATIONAL COMPLEXITY ANALYSIS

For each hyperparameter candidate, CV-imputation repeats the following procedure:

1. **Partition step.** Constructing and randomly partitioning the edge set $\{(v_i, v_j) : 1 \leq i < j \leq n\}$ into $K$ folds touches each edge once, so the cost is $O(n^2)$.

2. **Imputation step.** For a fixed fold $k$, constructing the training matrix $\mathbf{A}^{\text{train}}$ by masking and imputing the edges in $S_k$ visits $|S_k| \asymp n^2/K$ entries, so the cost is $O(n^2/K)$ per fold and $O(n^2)$ over all $K$ folds.

3. **Training step.** Fitting model $M$ to $\mathbf{A}^{\text{train}}$ has cost $C_{\text{estim}}(n)$ per fold, hence $O(K\,C_{\text{estim}}(n))$ over all folds.

4. **Debiasing step.** In fold $k$, the debiased predictions are computed only for the edges in validation set $S_k$, because these are the entries involved in the fold-$k$ validation loss. Since $|S_k| \asymp n^2/K$, the cost is $O(n^2/K)$ per fold and $O(n^2)$ overall across all folds.

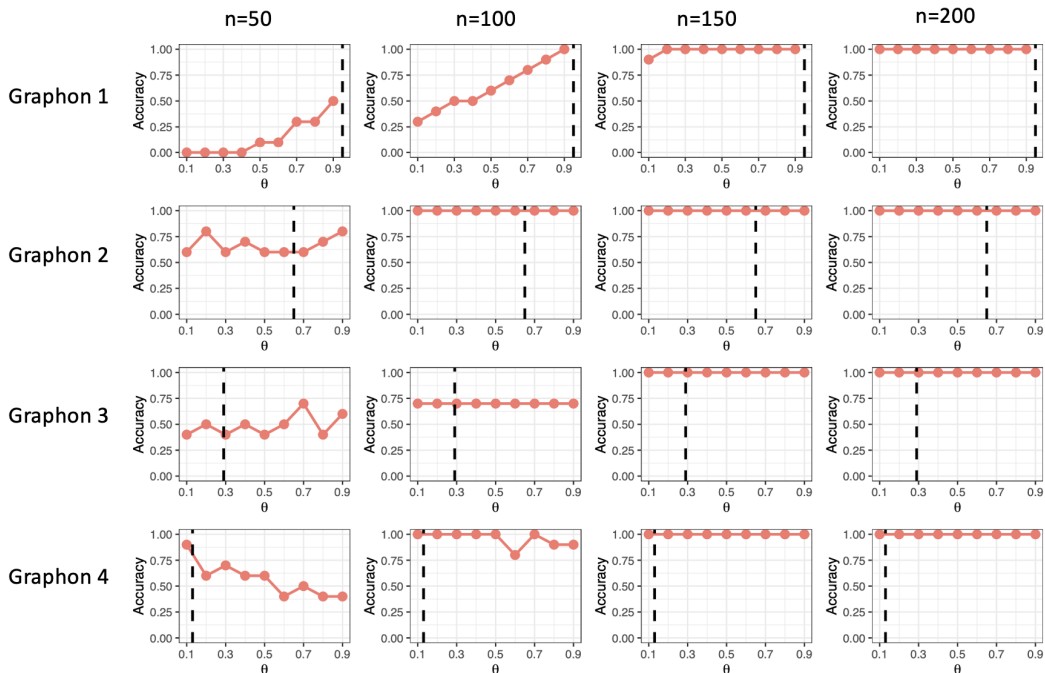

Figure S.2: The plots illustrate selection accuracy—defined as the percentage of instances where our method correctly identifies the optimal candidate model—against $\theta$ for sample sizes $50, 100, 150$, and $200$ (arranged from left to right) and Graphon models 1 to 4 (organized from top to bottom). The approximate network densities of 0.9, 0.6, 0.3, and 0.1 for Graphons 1 to 4 are represented by black dashed lines. It is evident that using the approximate density as an estimate for $\theta$ consistently achieves competitive selection accuracy. Furthermore, the influence of different choices of $\theta$ is most noticeable when $n$ is relatively small, such as 50 or 100.

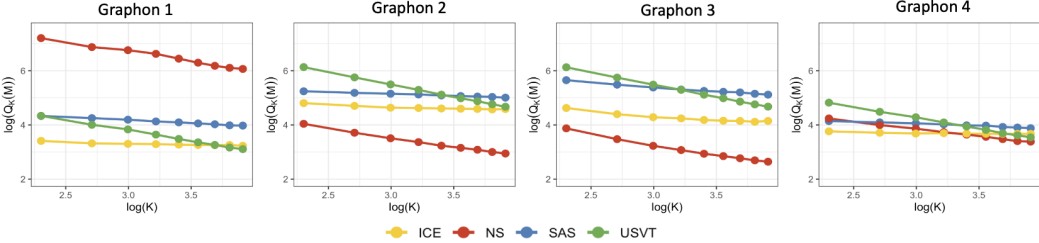

Figure S.3: K=10 to 50 for 100-node network generated from four different graphon functions listed in Figure 2. These results are averaged with 100 replicates.

5. **Validation score.** For each fold $k$, the validation error is computed by comparing the held-out edges $a_{ij}$, $(i,j) \in S_k$, with their debiased predictions. Since $|S_k| \asymp n^2/K$, the cost is $O(n^2/K)$ per fold and $O(n^2)$ across all folds.

Therefore, if we have $|\mathcal{M}|$ hyperparameters to evaluate, the total cost of CV-imputation for hyperparameter tuning is

$$O(|\mathcal{M}| \cdot (KC_{estim}(n) + n^2))$$

ECV follows the same outer loop structure with $K$ folds and identical partitioning. The critical difference lies in step (2): while our method performs imputation only, ECV performs both imputation (in a different way, i.e., masking $S_k$ to all zero) and matrix completion on the masked $n \times n$ matrix to produce a completed surrogate before estimation. This matrix completion step is computationally heavy, as it must operate on the full matrix and is repeated once per fold. The subsequent estimation

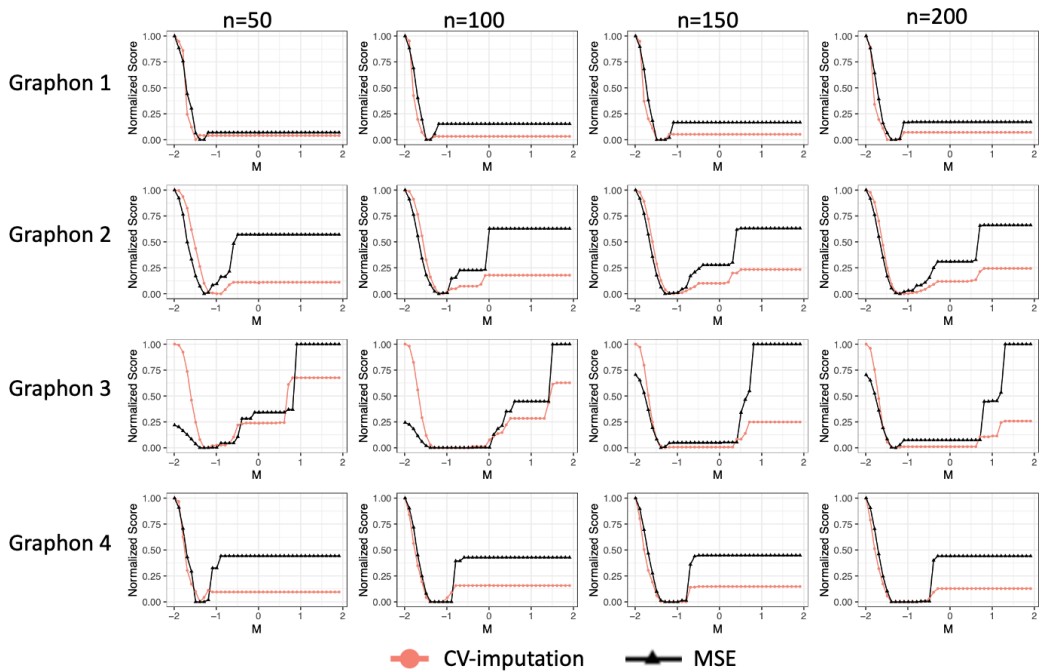

Figure S.4: Plotted here are the scores for CV-imputation (red) and MSE (black) under varying values of the tuning parameters for the USVT method. We vary the neighborhood size parameter $M$ from -2 to 2 with increments of 0.1, while the number of nodes $n$ ranges from 50 to 200 with an increment of 50. Each row corresponds to a specific graphon function listed in Figure 2.

and validation steps follow the same computational order as described above, except that ECV omits the de-biasing step. Therefore, ECV's total complexity is $O(|\mathcal{M}| \cdot (KC_{estim}(n) + KT_{\mathrm{mc}}(n)))$.

## S.9 DISCUSSION OF CONDITION 1

First, we discuss how Condition 1 implicitly depends on the graphon function $f$, $w_k$, and $\theta$.

**Dependence on the graphon function $f$.** While our framework does not explicitly impose assumptions on the graphon function $f$, the properties of $f$ play a fundamental and implicit role through Condition 1.

Condition 1 requires the estimator to be stable, i.e., imputing a small fraction $1/K$ of edges does not drastically change the probability estimate. This stability requirement can only be satisfied when $f$ satisfy the structural assumptions underlying the chosen graphon estimation method. If $f$ violates these assumptions, the estimator becomes unstable and Condition 1 fails to hold.

For example, the neighborhood smoothing method requires $f$ to be piecewise-Lipschitz continuous so that empirical row distances $\|A_i. - A_j.\|$ concentrate around their population counterparts. This smoothness condition ensures that neighborhood sets are stable and that perturbing only a small fraction of edges (as occurs in each CV fold) alters the estimator only mildly. In contrast, if $f$ were highly irregular or discontinuous, row distances would become unstable and small perturbations to the adjacency matrix could lead to large changes in the neighborhood structure, causing neighborhood smoothing to violate Condition 1.

**Dependence on $w_k$ and $\theta$.** (1) Condition 1 implicitly depends on the validation fraction $w_k \approx 1/K$. Since the perturbation size is proportional to $|S_k|$, a smaller $w_k$ (larger $K$) implies a smaller perturbation to the adjacency matrix. Consequently, it is intuitively easier for an estimator to maintain stability, potentially leading to a larger $\alpha$ or a smaller constant factor in the bound. (2) Condition

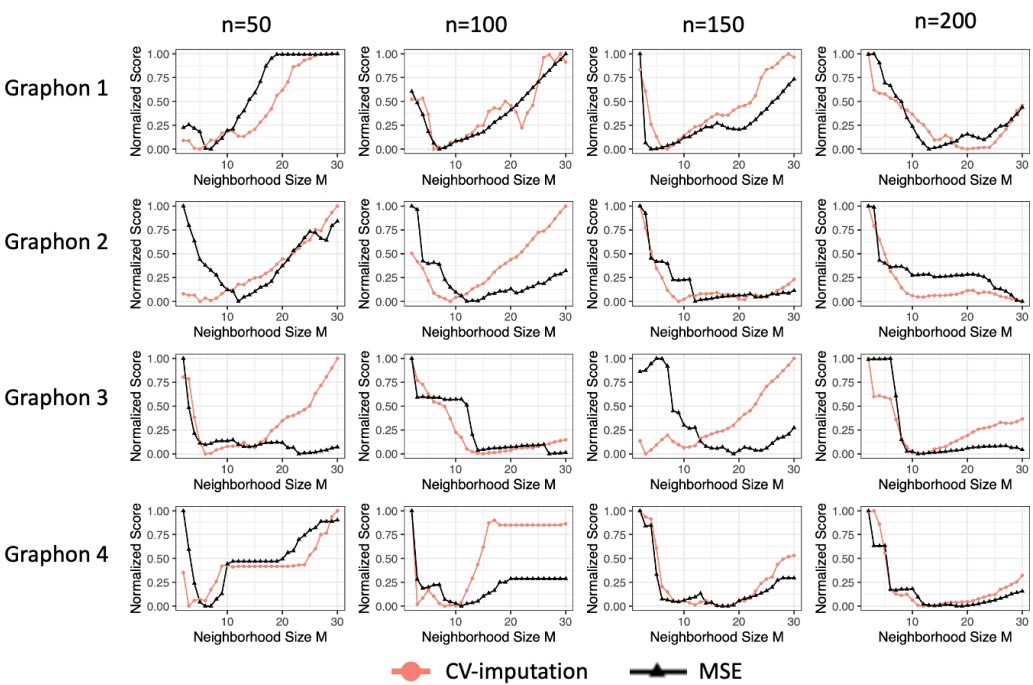

Figure S.5: Plotted here are the scores for CV-imputation (red) and MSE (black) under varying values of the tuning parameters for the SAS method. We vary the neighborhood size parameter $M$ from 2 to 30 with increments of 1, while the number of nodes $n$ ranges from 50 to 200 with an increment of 50. Each row corresponds to a specific graphon function listed in Figure 2.

1 also implicitly depends on $\theta$. For the estimator to remain stable (satisfying Condition 1 ), the imputation process should not introduce a systematic distributional shift between $\mathbf{A}$ and $\mathbf{A}^{\text{train}}$. Choosing $\theta$ close to the average network density minimizes the expected mean squared difference between the observed and imputed edges, thereby minimizing the "shock" to the estimator.

Second, we outline sufficient conditions and specific examples where Condition 1 holds.

**Sufficient Condition**: A sufficient condition for Condition 1 to hold with $\alpha = 1$ is that the graphon estimator $\hat{\mathbf{P}}(\mathbf{A})$ is Lipschitz continuous, i.e., $\|\hat{\mathbf{P}}(\mathbf{A}) - \hat{\mathbf{P}}(\mathbf{A}^{train})\|_F \leq L\|\mathbf{A} - \mathbf{A}^{train}\|_F$. Since $\mathbf{A}$ and $\mathbf{A}^{\text{train}}$ differ only on the validation set $S_k$, the squared difference is proportional to $|S_k| \asymp n^2/K$. Thus, $Q_K(M) \approx O(1/K)$, satisfying Condition 1 with $\alpha = 1$.

**Example A**: Erdős–Rényi Model. As mentioned in Section 4, the simple averaging estimator is a linear operator and trivially satisfies the Lipschitz condition.

**Example B**: Kernel Neighborhood Smoothing. Consider a kernel smoothing estimator defined by $\hat{P}_{ij}(\mathbf{A}) = \mathbf{W}(\mathbf{A}) \circ \mathbf{A}$ where $\circ$ is the hadamard product, and the $(i, k)$th entry of $\mathbf{W}(\mathbf{A})$ is the normalized weight $W_{ik} = \frac{d_{ik}}{\sum_{l=1}^{n} d_{il}}$, where $d_{ik} = K(\|A_{i\cdot} - A_{k\cdot}\|_2)$, and $K(\cdot)$ is a Lipschitz continuous kernel function (e.g., Gaussian kernel $e^{-x^2/h}$ with bandwidth $h$).

In this case, we have $\|\hat{\mathbf{P}}(M|\mathbf{A}) - \hat{\mathbf{P}}(M|\mathbf{A}^{train})\|_F^2 \leq \|\mathbf{W}(\mathbf{A}) \circ (\mathbf{A} - \mathbf{A}^{train})\|_F^2 + \|[\mathbf{W}(\mathbf{A}) - \mathbf{W}(\mathbf{A}^{train})]\mathbf{A}^{train}\|_F^2$, which is further bounded by $\|\mathbf{A} - \mathbf{A}^{train}\|_F^2$ and $\|\mathbf{W}(\mathbf{A}) - \mathbf{W}(\mathbf{A}^{train})\|_F^2$ since entries in $\mathbf{W}(\mathbf{A})$ and $\mathbf{A}$ are less than 1. By assuming the average of weights for any row is bounded away from zero, i.e., $\frac{1}{n} \sum_{l=1}^{n} K(\|A_{i\cdot} - A_{l\cdot}\|_2) \geq \delta > 0$, as $n \to \infty$, we can prove that $\mathbf{W}(\mathbf{A})$ is Lipschitz continous with respect to $\mathbf{A}$ given the lipschitz continuity of $K(\cdot)$. Therefore, the condition 1 can be satisifed with $\alpha = 1$ as well.

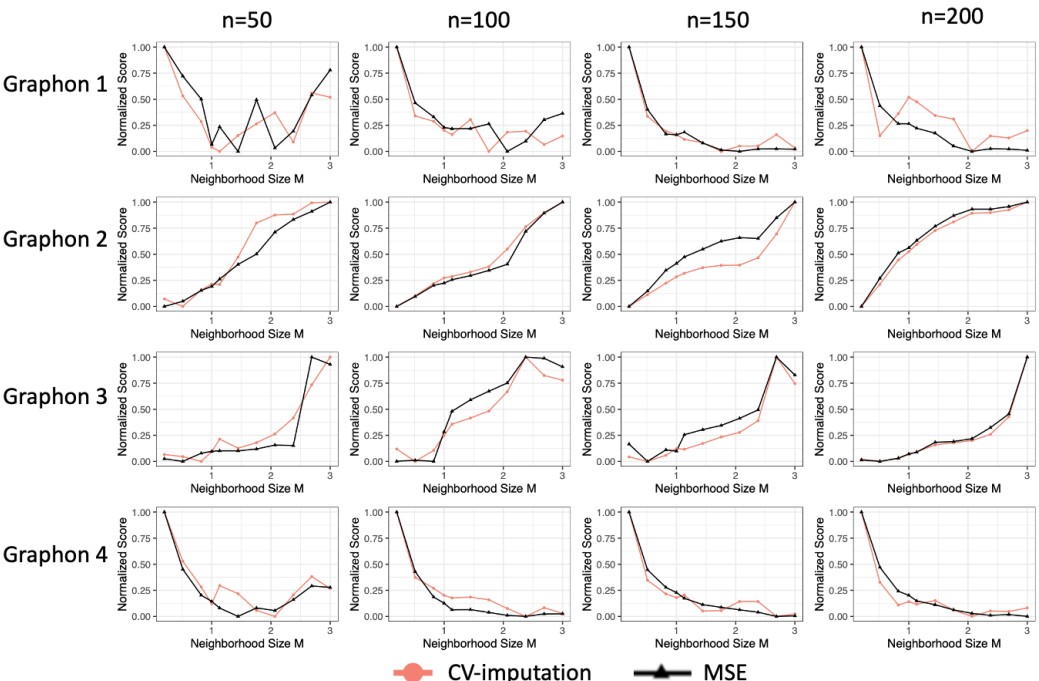

Figure S.6: Plotted here are the scores for CV-imputation (red) and MSE (black) under varying values of the tuning parameters for the ICE method. We vary the neighborhood size parameter $M$ from 0.2 to 3 with increments of 0.3, while the number of nodes $n$ ranges from 50 to 200 with an increment of 50. Each row corresponds to a specific graphon function listed in Figure 2.

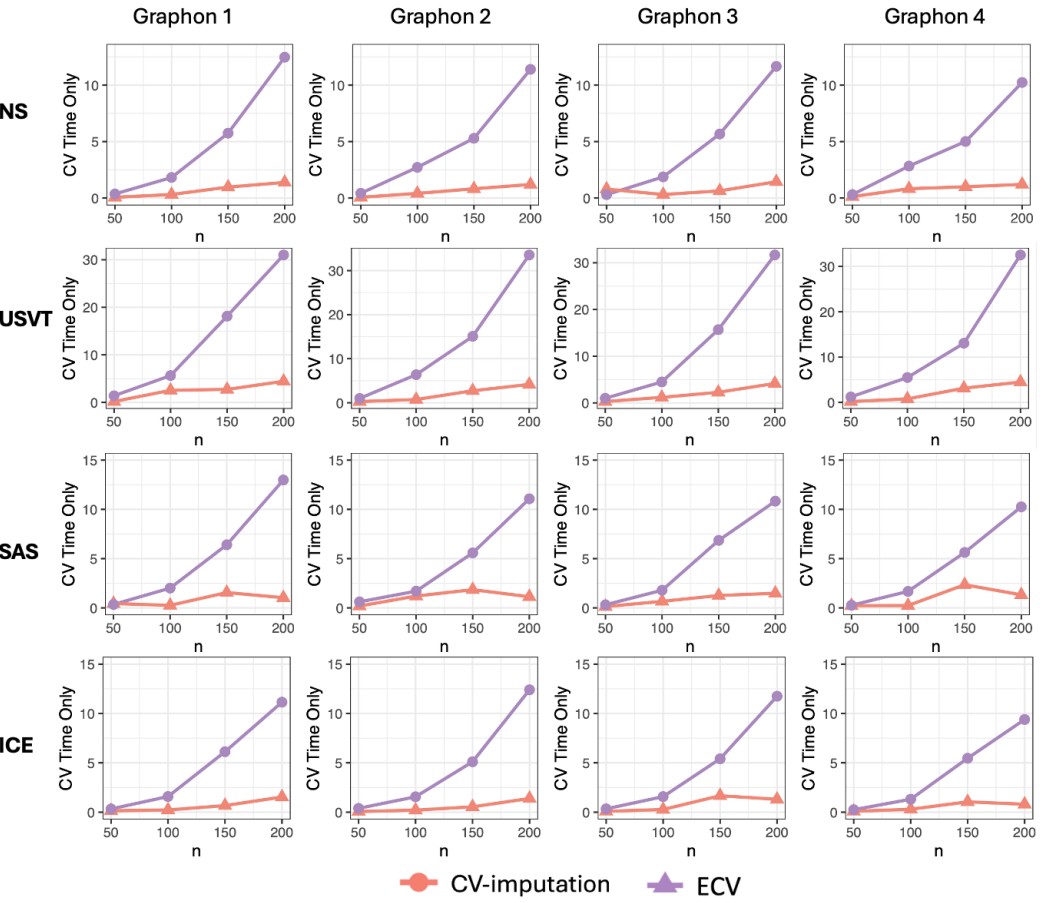

Figure S.7: The plots display the average cross-validation only computational time (in seconds) for Graphons 1 to 4 with $n \in 50, 100, 150, 200$ over 100 replicates, arranged from left to right. The panel from top to bottom corresponds to the NS method, the USVT method, the SAS method, and the ICE method, respectively.