# OpenReview forum: "Graphon Cross-Validation: Assessing Models on Network Data"
_ICLR.cc/2026/Conference — ICLR 2026 Poster_

### Official Review · Reviewer_8g5b · 2025-10-24

**Soundness:** 3
**Presentation:** 3
**Contribution:** 2
**Rating:** 4
**Confidence:** 4

**Summary:**

This paper addresses the problem of model selection in graphon estimation by proposing a new cross-validation method for selecting tuning parameters and estimation approaches. The authors prove that their cross-validation score is asymptotically aligned with the estimation error. Extensive numerical simulations and real-data analyses are provided.

**Strengths:**

1. I feel the proposed method quite smart. By randomly replacing a portion of the edges, the authors introduce a controllable bias, while obtaining edges that are (conditionally) independent of the training set for validation. The idea is interesting and provides insights for existing research.

2.	The authors show that minimizing $V_k(M)$ approximately corresponds to minimizing $L(M)$, which provides a theoretical justification for the proposed method.

3. The authors have provided sufficient numerical simulations and applied their method to real data examples.

4. I enjoyed reading the paper. The writing is clear and well-structured.

**Weaknesses:**

1. It is confused whether $v_i$ in Line 97 is a typo, since it is unusual for a node label to be a random variable. In addition, if $v_i$ is intended to be $\mu_i$, then $p_{ij}$ would be random, see my Questions 1 and 2.

2. It is unusual that the paper does not impose any assumptions on the graphon function. It is unclear what role the graphon function plays in their framework.

3. The proofs need to be written more rigorously. See my Question 2, 5.

**Questions:**

1. Could the authors clarify whether line 560 contains a type error? Should it perhaps be something like $P((v_i,v_j)\in S_k)P(b_{ij}=1, a_{i'j'}=1) + P((v_i,v_j)\notin S_k)P(a_{ij}=1,a_{i'j'}=1)?$ In addition, could the authors explain how the term $P(a_{ij}=1)P(a_{i'j'}=1)$ in line 561 is obtained? If I understand correctly, the edges $(i,j)$ and $(i',j')$ are correlated when $i=i'$ since they share the same $\mu_i$.

2. Line 597 could be made clearer. More precisely, the statement holds conditional on all $\mu_i$'s.

3. If all $\mu_i$'s are treated as deterministic, then it is unclear why the paper introduces the graphon function.

4.  It would be helpful if the authors could comment on the computational complexity of calculating $V_K(M)$, for example when $K\asymp n$.

5.  Could the authors clarify why Line 610 is correct? It seems that $S_k$ is random.


6. It would be helpful if the authors could provide more clarification on when  Condition 1 is satisfied. For instance, including a toy example or specifying sufficient conditions would make it clearer. In addition, could the authors comment on whether this condition depends on $w_k$ and $\theta$?

7. It seems that the theorem holds for any fixed $\theta$. What would happen if $\theta = 0$ or 1? In addition, in the appendix the authors set $\theta$ as a random variable, then how would this affect the validity of the theorem?


8. It would be helpful if the authors could provide a theoretical comparison between their method and ECV in terms of implementation time.

9. It would be interesting to know how the theoretical results behave when $p_{ij}$ tends to zero as $n$ increases.

10.  Some minor errors:

Line 158: $w_k\theta$: $w_k\theta 11^\top$.

Line 249, 251: five: four.

Line 573: equation equation 8: equation 8.

---

> ### Author Response · Authors · 2025-11-20
> **Thank you for your comments (1/3)**
>
> We thank the reviewer for the dedicated and insightful review. Please see below for our response.
>
> >  ***Comment:** It is confused whether $v_i$ in Line 97 is a typo, since it is unusual for a node label to be a random variable.*
>
>
> **Response:**  We apologize for the oversight. It is $\mu_i$ instead of $v_i$. We have corrected this typo.
>
>
> >  ***Comment:** It is unusual that the paper does not impose any assumptions on the graphon function. It is unclear what role the graphon function plays in their framework.*
>
> **Response:** We thank the reviewer for this insightful question. While our framework does not explicitly impose assumptions on the graphon function $f$, the properties of $f$ play a fundamental and implicit role through Condition 1.
>
>
> Condition 1 requires  the estimator to be stable, i.e., imputing a small fraction $1/K$ of edges does not drastically change the probability estimate. This stability requirement can only be satisfied when $f$ satisfy the structural assumptions underlying the chosen graphon estimation method. If $f$ violates these assumptions, the estimator becomes unstable and Condition 1 fails to hold.
>
> For example, the neighborhood smoothing method requires $f$ to be piecewise-Lipschitz continuous so that empirical row distances  $\|A_{i\cdot}-A_{j\cdot}\|$ concentrate around their population counterparts. This smoothness condition ensures that neighborhood sets are stable and that perturbing only a small fraction of edges (as occurs in each CV fold) alters the estimator only mildly.  In contrast, if $f$ were highly irregular or discontinuous, row distances would become unstable and small perturbations  to the adjacency matrix could lead to large changes in the neighborhood structure, causing neighborhood smoothing  to violate Condition 1.
>
>
>
> >  ***Comment:** Could the authors clarify whether line 560 contains a type error? how the term $P(a_{ij} = 1) P(a_{i j'} = 1)$ in line 561 is derived?  edges $(i,j)$ and $(i', j')$ are correlated.*
>
>
> **Response:** We thank the reviewer for this careful reading.
>
> (1) Clarification of line 560. The reviewer is correct, and we have revised as $P\big((v_i, v_j) \in S_k\big)P(b_{ij}=1, a_{i j'}=1) + P\big((v_i, v_j) \notin S_k\big)P(a_{ij}=1, a_{i j'}=1).$
>
> (2)  Independence of the terms in line 561. We agree that unconditionally, edges sharing a node are not independent under the graphon model. In our proof, however, all probability statements are taken conditional on the probability matrix $\mathbf{P}$ (equivalently, conditional on the latent variables $\{\mu_i\}$). Given $\mathbf{P}$, $a_{ij} \mid \mathbf{P} \sim \mathrm{Bernoulli}(p_{ij}), a_{i'j'} \mid \mathbf{P} \sim \mathrm{Bernoulli}(p_{i'j'}),$ and the edges are conditionally independent. Therefore, $P(a_{ij}=1,\, a_{i'j'}=1 \mid \mathbf{P})
> = P(a_{ij}=1\mid \mathbf{P})\, P(a_{i'j'}=1\mid \mathbf{P})$.
>
>
> We emphasize that this conditional independence structure is not specific to our framework; it is standard in the graphon literature. For example, [1] explicitly state that “$a_{ij}$’s are independent Bernoulli$(P_{ij})$ trials,” and [2] note that “given $\mu_i$, the variables $\{a_{ij}\}$ are independent.”
>
>
>
> [1] Zhang, Yuan, Elizaveta Levina, and Ji Zhu. "Estimating network edge probabilities by neighbourhood smoothing." Biometrika 104, no. 4 (2017): 771-783.
>
> [2]  Gao, Chao, Yu Lu, and Harrison H. Zhou. "RATE-OPTIMAL GRAPHON ESTIMATION." The Annals of Statistics (2015): 2624-2652.
>
> **Action**: We have revised the statement in Lemma 1 and its proof.
>
>
>
> >  ***Comment:** Line 597 could be made clearer. More precisely, the statement holds conditional on all $\mu_i$'s.*
>
> **Response:** We thank the reviewer for pointing out this.  The result indeed holds *conditional on all latent variables* $\{\mu_i\}$. In the graphon framework, this is equivalent to conditioning on the probability matrix $\mathbf{P}$, since each entry is given by $p_{ij} = f(\mu_i,\mu_j)$.
>
> To address the reviewer’s comment, we have made the conditioning explicit in the  revised manuscript: (1) Page 12 (Lemma S.1):   We now explicitly state that the result holds *conditional on the probability  matrix*  $\mathbf{P}$. (2) Page 13 (Proof of Lemma S.1): In the expected value calculation, we added the sentence  “where the expectation is conditional expectation with any given $\mathbf{P}$”.

---

> ### Author Response · Authors · 2025-11-20
> **Thank you for your comments (2/3)**
>
> >  ***Comment:** If all $\mu_i$'s are treated as deterministic, then it is unclear why the paper introduces the graphon function.*
>
> **Response:** We thank the reviewer for raising this point. However, conditional independence does not imply that $\{\mu_i\}$ is determinnistic. We clarify that we do *not* treat the latent variables $\{\mu_i\}$ as deterministic. In the graphon framework, the $\mu_i$'s are random latent positions, typically i.i.d. Uniform $[0,1]$. The randomness of $\mu$ is required to satisfy the property of exchangeability, i.e., the joint distribution of its edges remains unchanged when the nodes are arbitrarily permuted.
>
>
> Conditioning on $\mathbf{P}$ in proofs is a technical tool to handle the randomness separately (e.g., bound errors via concentration on independent Bernoullis, then average over the distribution of $\mu_i$).
>
> >  ***Comment:** It would be helpful if the authors could comment on the computational complexity of calculating $V_K(M)$, for example, when $K \asymp n$.*
>
> **Response:**  Thank you for your comment. The computational complexity of calculating the $V_K(M)$ score (as defined in Equation 7) is $O(n^2)$.
>
> Specifically, the formula for $V_K(M)$ is a sum over all $K$ validation sets. In aggregate, these $K$ sets perfectly partition the $\frac{n(n-1)}{2}$ unique node pairs. Consequently, once the fitted probabilities are available, the double summation $\sum_{k=1}^{K} \sum_{(v_i, v_j) \in \mathcal{S}_k}$ is simply a sum over all $O(n^2)$ node pairs. Thus, the computational complexity of computing $V_K(M)$ is $O(n^2)$, even when $K \asymp n$.
>
> >  ***Comment:**  Could the authors clarify why Line 610 is correct? It seems that $S_k$ is random.*
>
> **Response:** Thank you for pointing this out. Yes, $S_k$ is random. The original manuscript considers the randomness of $S_k$ implicitly. To address this, we have revised the appendix to make the conditioning explicit. We now apply the law of total variance and write $Var(V_K(M)- L_K(M)) = E[Var(V_K(M)- L_K(M) \mid \{S_k\})] + Var(E[V_K(M)- L_K(M) \mid \{S_k\}]).$ From Lemma S.1, $E[V_K(M)- L_K(M) \mid \{S_k\}] = \Lambda$, which does not depend on the partition $\{S_k\}$. Therefore, the second term equals to 0, and we only analyze the first term. Conditioned on the partition $\{S_k\}$, simple algebra yields $Var(V_K(M)- L_K(M) \mid \{S_k\})=(\frac{2}{n(n-1)})^2(\Delta_3+\Delta_4).$
>
> All subsequent variance calculations, including the definitions of $\Delta_3$ and $\Delta_4$, are therefore interpreted conditional on the realized partition $\{S_k\}$. After obtaining these conditional bounds, we then take the expectation with respect to the randomness of $\{S_k\}$, which yields the final variance expression, as shown in Line 717-730 in the revised mauscript.
>
>
>
>
>
>
> >  ***Comment:** More clarification on when Condition 1 is satisfied.*
>
>
> **Response:** Thank you for the insightful suggestion. Condition 1 essentially requires the estimator to be algorithmically stable with respect to the perturbation introduced by our imputation step. Below, we outline sufficient conditions and specific examples where this holds:
>
>
> 1. **Sufficient Condition**: A sufficient condition for Condition 1 to hold with $\alpha = 1$ is that the graphon estimator $\hat{\mathbf{P}}(\mathbf{A})$ is Lipschitz continuous, i.e., $\Vert\hat{\mathbf{P}}(\mathbf{A}) - \hat{\mathbf{P}}(\mathbf{A}^{train})\Vert_F \leq L \Vert\mathbf{A} - \mathbf{A}^{train}\Vert_F$. Since $\mathbf{A}$ and $\mathbf{A}^{\text{train}}$ differ only on the validation set $S_k$, the squared difference is proportional to $|S_k| \asymp n^2/K$. Thus, $Q_K(M) \approx O(1/K)$, satisfying Condition 1 with $\alpha=1$.
>
> **Example A**: Erdős–Rényi Model. As mentioned in the manuscript, the simple averaging estimator is a linear operator and trivially satisfies the Lipschitz condition.
>
> **Example B**: Kernel Neighborhood Smoothing. Consider a kernel smoothing estimator defined by $\hat{P}_{ij}(\mathbf{A}) = \mathbf{W}(\mathbf A)\circ \mathbf{A}$ where $\circ$ is the hadamard product, and the $(i,k)$th entry of $\mathbf{W}(\mathbf A)$ is the normalized weight
>
> $W_{ik} = \frac{d_{ik}}{\sum_{l=1}^n d_{il} }$, where $d_{ik}=Kernel(\Vert A_{i\cdot} - A_{k\cdot}\Vert_2)$, and $K(\cdot)$ is a Lipschitz continuous kernel function (e.g., Gaussian kernel $e^{-x^2/h}$ with bandwidth $h$).

---

> ### Author Response · Authors · 2025-11-20
> **Thank you for your comments (3/3)**
>
> In this case, we have $\Vert\hat{\mathbf{P}}(M|\mathbf{A})-\hat{\mathbf{P}}(M|\mathbf{A}^{train})\Vert_F^2 \leq \Vert\mathbf{W}(\mathbf A)\circ(\mathbf A-\mathbf A^{train})\Vert_F^2+\Vert[\mathbf{W}(\mathbf A)-\mathbf{W}(\mathbf A^{train})] \mathbf A^{train} \Vert_F^2$, which is further bounded by $\Vert\mathbf A-\mathbf A^{train}\Vert_F^2$ and $\Vert\mathbf{W}(\mathbf A)-\mathbf{W}(\mathbf A^{train})\Vert_F^2$ since entries in $\mathbf{W}(\mathbf A)$ and $\mathbf{A}$ are less than 1. By assuming the average of weights for any row is bounded away from zero, i.e., $\frac{1}{n}\sum_{l=1}^n K(\Vert A_{i\cdot} - A_{l\cdot}\Vert_2) \geq \delta >0$, as $n \to \infty$, we can prove that $\mathbf{W}(\mathbf A)$ is Lipschitz continous with respect to $\mathbf A$ given the lipschitz continuity of $K(\cdot)$. Therefore, the condition 1 can be satisifed with $\alpha=1$ as well.
>
> 2. **Another Example**: Maximum Likelihood Estimation ($\alpha = 2$). As discussed in General Response \#2, if the data is generated by an inner-product functin $f(\mu_i, \mu_j) = \mu_i \mu_j$, the Maximum Likelihood Estimator (MLE) under boundedness constraints satisfies Condition 1 with an even faster rate of $\alpha=2$.
> 3. **Dependency on $w_k$ and $\theta$**. (1) Condition 1 implicitly **depends** on the validation fraction $w_k \approx 1/K$. Since the perturbation size is proportional to $|S_k|$, a smaller $w_k$ (larger $K$) implies a smaller perturbation to the adjacency matrix. Consequently, it is intuitively easier for an estimator to maintain stability, potentially leading to a larger $\alpha$ or a smaller constant factor in the bound. (2)  Condition 1 also implicitly **depends** on $\theta$.  For the estimator to remain stable (satisfying Condition 1), the imputation process should not introduce a systematic distributional shift between  $\mathbf{A}$ and $\mathbf{A}^{\text{train}}$. Choosing $\theta$ close to the average network density minimizes the expected mean squared difference between the observed and imputed edges, thereby minimizing the "shock" to the estimator.
>
>
>
> >  ***Comment:** It seems that the theorem holds for any fixed $\theta$. What would happen if $\theta = 0$ or $1$? In addition, in the appendix the authors set $\theta$ as a random variable, then how would this affect the validity of the theorem?*
>
> **Response:** We thank the reviewer for this insightful comment.
>
> (1) **Effect of  $\theta$**. The reviewer is correct that Theorem 1 holds theoretically for any fixed $\theta \in [0,1]$, provided that the stability assumption (Condition 1) is met. However, the choice of $\theta$ significantly impacts whether Condition 1 hold.
>
>
> For example, if the true network is sparse (where most $a_{ij}=0$) and we set $\theta=1$, we introduce a systematic bias and a large perturbation variance. This "dense noise" can destroy the sparsity structure of the training network.  Conversely, setting $\theta=0$ in a dense network would systematically underestimate relationships.
>
>
> **(2) Clarification on $\theta$ as a fixed hyperparameter.** In practice, we recommend setting $\theta$  using a value close to the empirical graph density. This is just a practical suggestion for hyperparameter selection, not a part of the generative model. This is analogous to selecting a bandwidth in kernel density estimation based on a "rule of thumb" (like Silverman’s rule).
>
>
>
> >  ***Comment:** It would be helpful if the authors could provide a theoretical comparison between their method and ECV in terms of implementation time.*
>
> **Response:**  Thank you for your comment. Please see our **General Response \#1** to find the  asymptotic computational complexity comparison between our CV-imputation method and ECV.
>
>
>
>
>
>
> >  ***Comment:**  It would be interesting to know how the theoretical results behave when $p_{ij}$ tends to zero as $n$ increases.*
>
> **Response:** Thank you for the insightful comment. This case corresponds to the generalized sparse graphon setting $p_{ij} = \rho_n f(\mu_i, \mu_j)$, where $\rho_n$ decays as $n$ increases.  We refer the reviewer to **General Response \#2**, where we provide (i) empirical validation showing that CV-imputation continues to perform well (**Table R2**), with $\rho_n = \sqrt{\frac{log n}{n}}$, and (ii) a discussion of the theoretical extension and the challenges of adapting Theorem 1 to sparsity-adjusted rates. Extending the theory to fully sparse regimes is an important and promising direction for future work.
>
>
>
>
> >  ***Comment:** Some minor errors: Line 158: $w_k \theta$: $w_k \theta 11^{\top}$. Line 249, 251: five: four. Line 573: equation equation 8: equation 8.*
>
> **Response:** Thank you for your comments, we have corrected these typos.

---

> > ### Comment · Reviewer_8g5b · 2025-11-21
> >
> > Thank you for your clarification and revision of the paper. I encourage the authors to add the explanation into the revised version, particularly the clarification on Condition 1 (If I understand correctly, it imposes implicit constraints on the estimation method, the smoothness of graphon, and parameters $w_k ,\theta$). I have adjusted my score accordingly.

---

> > > ### Author Response · Authors · 2025-11-21
> > > **Thank you**
> > >
> > > Thank you very much for your positive reassessment of our manuscript. Following your suggestions, we have added the clarification of Condition 1 in the revised manuscript (the updated version has been uploaded).
> > >
> > > Page 5: We have added one sentence, "Further discussion of Condition 1 can be found in Section S.10".
> > >
> > > Pages 22-23: We have incorporated the discussion of Condition 1 in Section S.10 in the Appendix, including how Condition  1 implicitly depends on  $f$, $w_k$, and $\theta$, and the sufficient condition for Condition 1 to hold.

---

### Official Review · Reviewer_9pGj · 2025-10-29

**Soundness:** 4
**Presentation:** 4
**Contribution:** 3
**Rating:** 8
**Confidence:** 4

**Summary:**

This paper introduces an improved cross-validation method with random imputation, which is unbiased and efficient compared to existing methods including edge sampling approaches and the matrix completion method proposed by Li et al. (2020). The paper compares against these baselines across multiple graphon estimation methods, demonstrating effectiveness in terms of both accuracy and computational efficiency. The authors provide theoretical justification showing that their cross-validation score is asymptotically correct. I found that this paper is strong and provides a comprehensive analysis based on both theory and numerical evidence. The case studies are insightful and demonstrate practical applicability of the proposed method.

**Strengths:**

1. The authors effectively articulate the fundamental challenge in applying cross-validation to network data, particularly how traditional edge sampling destroys network topology.

2.  The random imputation strategy is quite simple and very effective. I could not think of any simpler way.

3. The paper provides extensive experiments across multiple graphon models (varying in density and rank properties) and estimation methods (NS, SAS, USVT, ICE). Section 6's case studies are particularly compelling.

4. Unlike ECV which requires low-rank assumptions and only has theoretical guarantees for specific models (SBM, RDPG), CV-imputation works for the general graphon model class

**Weaknesses:**

1. I could not follow the theoretical justification precisely, but I get the intuition behind it which is that the training and test data are independent of each other. Therefore, it doesn't change the distribution of the edge probabilities. It is worthwhile to note that this is valid only when the presence orabsense of an edge is independent of that of other edges.

2.  While the paper tests networks up to approximately 2,600 nodes, many real-world networks contain tens or hundreds of thousands of nodes, yet no discussion addresses computational or memory requirements at such scales. I understand to some extent because the number of pairs for increases quadratically with respect to the number of nodes. But it is nicer to discuss a solution to scale up the method.

**Questions:**

I am unclear why the computational time differs across estimation methods (Figure 3), since the cross-validation procedure itself should be independent of the estimation method being evaluated. I suspect the authors calculated the total CPU time, which includes both the cross-validation procedure and the estimation method execution, rather than isolating the cross-validation overhead alone. This make it unclear the actual speedup specifically attributable to CV-imputation compared to ECV. The authors should clarify what operations are included in their reported CPU times and, ideally, provide a breakdown separating cross-validation overhead from estimation costs.

---

> ### Author Response · Authors · 2025-11-20
> **Thank you for your comments**
>
> We thank the reviewer for the dedicated and insightful review. Please see below for our response.
>
> >  ***Comment:** I could not follow the theoretical justification precisely, but I get the intuition behind it which is that the training and test data are independent of each other. Therefore, it doesn't change the distribution of the edge probabilities. It is worthwhile to note that this is valid only when the presence or absence of an edge is independent of that of other edges.*
>
> **Response:** We thank the reviewer for this helpful comment. The reviewer is correct that our justification relies on an independence structure, but we clarify the precise form needed.
>
> Lemma 1 does *not* require unconditional independence of all edges. Instead, it uses the standard assumption from the graphon framework that edges are *conditionally independent given the latent probability matrix*  $P = (p_{ij})$. That is, conditional on **P**, edges $a_{ij} \mid \mathbf{P} \sim \mathrm{Bernoulli}(p_{ij})$ are mutually independent.
>
>
> Under this conditional independence, the masked training network and the validation set are independent given **P**, which is precisely the content of Lemma 1.
>
> **Action:** In the revised manuscript, we have revised the discussion of Lemma 1 to make this point explicit.
>
>
>
> >  ***Comment:** Discussion  on computational or memory requirements, and scalability issue.*
>
> **Response:** We appreciate the reviewer’s thoughtful comment regarding scalability.
>
> **Computational and memory requirements.** We have provided an explicit asymptotic computational complexity analysis of our CV-imputation method, please see General Response \#1 to find details. Briefly speaking, to evaluate a set of $|\mathcal{M}|$ candidate hyperparameters on an $n$-node network using $K$-fold cross-validation, the total computational complexity of our CV-imputation method is  **$O(|\mathcal{M}| \cdot (KC_{estim}(n) + n^2))$**, where $C_{\text{estim}}(n)$ denotes the cost of the specific graphon estimator (e.g., NS, ICE). Existing graphon estimation methods usually have $C_{estim}$ greater than $n^2$. So the cost is mainly dominated by the graphon estimation method itself rather than by CV-imputation itself.
>
>
> In terms of memory, the main memory cost is storing the adjacency and the full predicted probability matrix for the estimator, which requires $O(n^2)$. We do not need to keep $K$ copies of the adjacency; a single $n \times n$ buffer can be reused, with $n^2/K$ entries temporarily imputed for fold and then reset. The fold assignments can be stored explicitly as fold indices $O(n^2)$. Thus the memory of CV-imputation is dominated by $O(n^2)$.
>
>
> **Scaling to very large networks.** For very large networks, a practical way to scale CV-imputation is to combine it with network subsampling: (1) extract a structurally representative subgraph using network sampling methods (e.g., Metropolis-Hastings random walk [1], curvature-based sampling [2]), (2) apply CV-imputation on this subgraph to tune hyperparameters, and (3) fit the full network using the selected hyperparameter.
>
> **Action**: Page 4 in the revised manuscript, we have added a paragraph for such discussion.
>
> References:
>
> [1] Hu, P., & Lau, W. C. (2013). A survey and taxonomy of graph sampling. arXiv preprint arXiv:1308.5865.
>
> [2] Wu, S., Cheng, H., Cai, J., Ma, P., & Zhong, W. (2023, April). Subsampling in large graphs using ricci curvature. In International Conference on Learning Representations.
>
> >  ***Comment:** Why the runtime in Figure 3 differs across estimators?*
>
> **Response:** Thank you for this helpful comment. In Figure 3, we report the **end-to-end CPU time** required to complete the hyperparameter–tuning pipeline. This includes (i) constructing folds and masks for cross-validation, (ii) fitting the graphon estimator on each training set, (iii) computing predictions/imputations on validation edges, and (iv) evaluating and aggregating the validation loss over the hyperparameter grid. Because the underlying estimators (NS, USVT, SAS, ICE) have different computational complexity, their end-to-end tuning times naturally differ.
>
> To directly address your concern, we have **added a new figure in the Appendix (Figure S.7)** that isolates the cross-validation overhead by subtracting the graphon estimation time from the total runtime. This figure reports the CPU time of the cross-validation procedure alone for ECV and CV-imputation. The results show that (1) the runtime is similar across estimators, (2) CV-imputation is consistently faster than ECV at the level of the cross-validation procedure itself.
>
> **Action**: (1) On Pages 5–6 in the revised manuscript, we have revised the main text to explicitly state which components of the pipeline are included in the CPU times reported in Figure 3. (2) On page 21, we have added Figure S.7 to show the time for cross-validation only.

---

> > ### Comment · Reviewer_9pGj · 2025-11-21
> > **Response**
> >
> > I appreciate authors for responding to my concerns and I'm satisfied with the authors' response.

---

> > > ### Author Response · Authors · 2025-11-21
> > > **Thank you**
> > >
> > > We sincerely appreciate your careful review and constructive feedback!

---

### Official Review · Reviewer_NTvL · 2025-11-02

**Soundness:** 3
**Presentation:** 3
**Contribution:** 2
**Rating:** 4
**Confidence:** 4

**Summary:**

This paper proposes a new cross-validation approach for graphon estimation, introducing a random-imputation strategy to handle dependency structures in networks and to tune hyperparameters from a single observed graph. The authors argue that existing edge-sampling methods lead to bias by degrading network connectivity, and they provide a careful theoretical treatment to show consistency of their approach. The work is technically sound and well written, but the contribution represents a relatively narrow conceptual advance within the specific context of graphon models rather than a broader methodological step forward for network machine learning. The empirical results, while generally positive, do not convincingly demonstrate consistent or substantial gains over edge-based cross-validation (ECV). The paper would be strengthened by a deeper analysis of when and why ECV fails in practice, clearer discussion of how the proposed method could generalize beyond the graphon setting, and a more comprehensive evaluation showing robust and meaningful performance improvements across a wider range of network types.

**Strengths:**

The paper addresses a legitimate technical challenge: how to form cross-validation sets in under dependence in networks. The proposed CV-imputation scheme is clearly described, mathematically justified, and accompanied by consistency proof. The implementation appears efficient, and the experiments consider several graphon estimators (NS, SAS, USVT, ICE), as well as both synthetic and real-world graph datasets. The manuscript is generally clear and thorough.

**Weaknesses:**

(1)	Scope and generality. The method applies specifically to graphon models and depends on smoothness and exchangeability assumptions. It is unclear how the proposed theoretical framework or random-imputation idea would extend to broader classes of network models (e.g., latent-space, temporal, or sparsified networks). Thus the paper’s claims of general applicability thus seem overstated.
(2)	Empirical impact. The experimental gains over ECV are modest. In Table 1, SAS and ICE show no meaningful improvement, suggesting that differences arise mainly from estimator robustness rather than from the proposed validation method. In real-world datasets, results are essentially tied on three of four networks. Only the drug–disease network shows a visible advantage, which is not analyzed in depth. Without more in-depth analysis demonstrating the impact of ECV bias, it is difficult to conclude that the differences are practically important.
(3)	Efficiency gains. Although Figure 3 shows a computational speed-up relative to ECV, the paper does not provide an asymptotic complexity comparison. Moreover, the statistical efficiency results in Figure 5 are averaged across all graphon models and may be dominated by the weaker performance of NS and USVT, making it hard to assess whether the gains are consistent or meaningful across settings.

**Questions:**

See weaknesses.

---

> ### Author Response · Authors · 2025-11-20
> **Thank you for your comment (1/2)**
>
> We thank the reviewer for the dedicated and insightful review. Please see below for our response.
>
> >  ***Comment:** Extension to broader classes of network models (e.g., latent-space, temporal, or sparsified networks).*
>
> **Response:**  Thank you for raising this important point.
>
> First, we clarify that the graphon model is not a "specific" model but a highly general, non-parametric framework. As noted in the literature [1], it encompasses many existing network models as special cases: (1) Erdős-Rényi (ER) model: setting the graphon function $f$ to be a constant recovers the ER model. (2) Stochastic Block Models (SBM): Setting the graphon function $f$ to be piecewise-constant recovers the SBM. (3) Latent Space Models: These models assume nodes have latent positions $\mu_i \in \mathbb{R}^d$, and connection probabilities usually depend on inner product or Euclidean distance of latent coordinates. In fact, Graphon 4 ($f = \mu_i \mu_j / 2$) in our paper is a special latent-space model with $d=1$.
>
>
> Second, in our **General Response \#2**, we provided new simulations and a theoretical extensions discussion.  Briefly, we conducted additional simulations under two representative non-classical settings: a latent-space model and a generalized sparse graphon model (Tables R1–R2). In both cases, CV-imputation achieves near-oracle performance and consistently outperforms ECV.  Theorem 1 is more complicated, depending on Condition 1, which requires case-by-case discussion for each estimation method. We showed in General Response \#2 that Condition 1 can be verified for a maximum-likelihood estimator for a latent-space model, allowing Theorem 1 to extend directly to this setting. For sparse graphon models, we explain how the theory would need to incorporate the sparsity level and view this as a promising direction for future work.
>
> Third,  temporal or dynamic networks violate our key assumption because edges across time are dependent by construction, training, and validation sets would no longer be independent even after random imputation. This edge-independence makes Lemma 1 fail.
>
> [1] Xu, Jiaming. "Rates of convergence of spectral methods for graphon estimation." International Conference on Machine Learning. PMLR, 2018.
>
> >  ***Comment:** The experimental gains over ECV are modest. In Table 1, SAS and ICE show no meaningful improvement. In real-world datasets, results are essentially tied on three of four networks.*
>
>
> **Response:** Thank you for this thoughtful comment.
>
> **Clarification on SAS and ICE Performance.** SAS and ICE are inherently more insensitive to hyperparameter choices due to their structural assumptions: SAS leverages degree sorting and total variation minimization, which implicitly regularizes against over-smoothing, while ICE iteratively refines probabilities with less sensitivity to initial neighborhood sizes. In contrast, neighborhood smoothing and universal singular value thresholding (USVT) are more parameter-dependent, leading to larger relative improvements with CV-imputation.  The modest gains observed for SAS and ICE therefore reflect the intrinsic stability of these estimators rather than any limitation of CV-imputation—no tuning approach can produce large improvements when the underlying method is already insensitive to its hyperparameters.
>
>
> To confirm this explanation, we computed the Oracle MSE (best possible MSE attainable over the entire grid of tuning parameters). As shown in the table below, CV-imputation is nearly identical to the Oracle for SAS and ICE.
>
> Table R3: Mean $\pm$  standard deviation of the oracle MSE across 100 replicates (multiplied by 100 for readability), with $n=200$.
>
> | Method        | Graphon 1        | Graphon 2        | Graphon 3        | Graphon 4        |
> |---------------|------------------|------------------|------------------|------------------|
> | Oracle (SAS)  | 1.68 ± 0.09      | 8.18 ± 0.18      | 12.55 ± 0.21     | 1.45 ± 0.09      |
> | Oracle (ICE)  | 0.30 ± 0.04      | 2.52 ± 0.22      | 0.48 ± 0.04      | 0.80 ± 0.05      |
>
>
> **Clarification on Real-World Results in Table 2**. Results are not "essentially tied" on three networks: Paired Wilcoxon rank-sum tests on 100 replicates confirm significant improvements for PolBlog (AUC 0.88 $\pm$ 0.01 vs. 0.80 $\pm$ 0.02, p<0.01) and NetSci (0.72 $\pm$ 0.01 vs. 0.70 $\pm$ 0.01, p<0.05), with comparable performance on Yeast. Combined with 4-5x speedups (lower panel of Table  2), this underscores practical advantages for large graphs.
>
>
> **Additional Clarification on Empirical Impact**. We emphasize that CV-imputation's gains are substantial where tuning matters most. In Table 1, it reduces errors by 2× to 10× over ECV for NS and USVT across graphons, reflecting our method's effectiveness. Figure 5's method-selection results show even stronger advantages: CV-imputation achieves near-100% accuracy vs. ECV's ~25% for Graphons 1–2 at $n$=100, highlighting meaningful improvements in method selection.

---

> > ### Author Response · Authors · 2025-11-20
> > **Thank you for your comment (2/2)**
> >
> > >  ***Comment:** Asymptotic complexity comparison.*
> >
> > **Response:** Thank you for raising this important point. We have provided an explicit asymptotic comparison between our CV-imputation method and ECV, please see **General Response \#1** to find details.
> >
> >
> >
> > >  ***Comment:** Moreover, the statistical efficiency results in Figure 5 are averaged across all graphon models and may be dominated by the weaker performance of NS and USVT, making it hard to assess whether the gains are consistent or meaningful across settings.*
> >
> > **Response:** Thank you for raising this point. To clarify, Figure 5 does not report averages of the hyperparameter-tuning task across the four estimation methods (NS, USVT, SAS, ICE), nor does it average across graphon models. Instead, it evaluates a different task: **method selection**. We clarify the distinction below.
> >
> > 1. Hyperparameter tuning (Figure 3 and Table 1). For each individual estimator, we select its optimal hyperparameter value. Figure 3 reports the computational time and Table 1 reports the resulting MSE for each method separately.
> > 2. Method selection (Figure 5). After each method has been tuned individually, we then perform a model selection step. In Figure 5, the top row  reports model-selection accuracy, the proportion of replications in which the selected (tuned) estimator matches the Oracle best estimator. The bottom row reports the additional time required for method selection, excluding the hyperparameter-tuning time already reported in Figure 3.
> >
> > **Action**: Page 7 and 8 in the revised manuscript, we have revised the caption and description for Figure 5 to clarify this.

---

> > > ### Author Response · Authors · 2025-11-25
> > > **Follow up on rebuttal**
> > >
> > > Dear Reviewer,
> > >
> > >
> > > Thank you again for taking the time to provide constructive feedback on our paper. It would be very helpful if you could kindly let us know whether our responses address all your concerns or if you have any additional comments. We look forward to hearing from you.
> > >
> > >
> > > Best,
> > >
> > > Authors

---

### Official Review · Reviewer_Q27V · 2025-11-03

**Soundness:** 3
**Presentation:** 3
**Contribution:** 3
**Rating:** 6
**Confidence:** 2

**Summary:**

The paper proposes CV-imputation, a cross-validation method for selecting graphon-based models on network data. The method is model-agnostic, computationally efficient, and supported by theoretical guarantees of asymptotic consistency. Empirical results show that it outperforms or matches existing methods across various networks.

**Strengths:**

1. CV-imputation does not assume a specific form of the graphon, allowing it to be applied across diverse network structures without model restrictions.
2. The method avoids expensive singular value decomposition (SVD), reducing runtime and enabling scalability to large networks.
3. It is supported by a convergence result showing that its validation criterion aligns with mean squared error minimization in the asymptotic regime.

**Weaknesses:**

This work assumes the data comes from a graphon and the goal is to assess graphon-based estimators. However, would such a method be extended beyond this assumption? Discussion about the limitations of applicability would be a nice addition to the paper.

Additional discussion on the order complexity of the method vs the baselines would strengthen the paper.

**Questions:**

1. How sensitive is CV-imputation to violations of the graphon assumption? Are the theoretical guarantees still meaningful outside the graphon framework?

---

> ### Author Response · Authors · 2025-11-20
> **Thank you for your comments**
>
> We thank the reviewer for the dedicated and insightful review. Please see below for our response.
>
>
>
> >  ***Comment:** Additional discussion on the order complexity of the method vs the baselines would strengthen the paper.*
>
> **Response:**  Thank you for the insightful comment. Please see our **General Response \#1** to find the  asymptotic computational complexity comparison between our CV-imputation method and ECV.  Briefly, evaluating $|\mathcal{M}|$ hyperparameters with CV-imputation has complexity $O(|\mathcal{M}| \cdot (KC_{estim}(n) + n^2))$, whereas ECV requires $O(|\mathcal{M}| \cdot   (KC_{estim}(n) + KT_{\text{mc}}(n)))$, with $C_{\text{estim}}(n)$ denoting the estimator cost and $T_{\text{mc}}(n)$ the matrix-completion cost. When matrix completion is implemented via a full SVD, we have $T_{\mathrm{mc}}(n) = O(n^3)$, and even with truncated SVD, $T_{\mathrm{mc}}(n)$ remains asymptotically larger than $n^2$, thus our method is faster than ECV.
>
>
> >  ***Comment:** How sensitive is CV-imputation to violations of the graphon assumption? Are the theoretical guarantees still meaningful outside the graphon framework?*
>
> **Response:**  We thank the reviewer for raising this important question. A detailed discussion, including new simulations and theoretical extensions, is provided in  our **General Response \#2**.
>
>
> Briefly, we conducted additional simulations under two representative non-classical settings: a latent-space model and a generalized sparse graphon model (Tables R1–R2). In both cases, CV-imputation achieves near-oracle performance and consistently outperforms ECV.  Theorem 1 is more complicated, depending on Condition 1, which require case-by-case discussion for each estimation method. We showed in General Response \#2 that Condition 1 can be verified for a maximum-likelihood estimator for latent-space model, allowing Theorem 1 to extend directly to this setting. For sparse graphon models, we explain how the  theory would need to incorporate the sparsity level  and view this as a promising  direction for future work.

---

> > ### Author Response · Authors · 2025-11-25
> > **Follow up**
> >
> > Dear Reviewer,
> >
> > Thank you again for taking the time to provide constructive feedback on our paper. It would be very helpful if you could kindly let us know whether our responses address all your concerns or if you have any additional comments. We look forward to hearing from you.
> >
> > Best,
> > Authors

---

### Author Response · Authors · 2025-11-20
**General Responses (1/3)**

We thank all reviewers for their thoughtful and constructive feedback. We are encouraged that they found the paper clear and well written (Reviewers 8g5b, NTvL), our proposed CV-imputation method “simple and very effective” (Reviewer 9pGj) and “quite smart” (Reviewer 8g5b), with theoretical soundness (Reviewers NTvL, Q27V, 8g5b, 9pGj) and strong empirical validation through "extensive experiments" (Reviewer 9pGj) and "sufficient numerical simulations" (Reviewer 8g5b).

### **1. Computational order complexity of the method**


**Response:** We thank reviewers for the insightful comment. To evaluate a set of $|\mathcal{M}|$ candidate hyperparameters on an $n$-node network using $K$-fold cross-validation, the total computational complexity of our CV-imputation method is  **$O(|\mathcal{M}| \cdot (KC_{estim}(n) + n^2))$**, while the competing ECV method has complexity **$O(|\mathcal{M}| \cdot   (KC_{estim}(n) + KT_{\text{mc}}(n)))$**. Here,  $C_{\text{estim}}(n)$ denotes the cost of the specific graphon estimator (e.g., NS, ICE), $T_{\text{mc}}(n)$ is the matrix completion complexity. When using full SVD for matrix completion, $T_{\text{mc}}(n)$ has $O(n^3)$, in which case, ECV has computation $O(|\mathcal{M}| \cdot   (KC_{estim}(n) + Kn^3))$.



*Detailed Complexity Breakdown:* For each hyperparameter candidate, CV-imputation repeats the following procedure:

(1) *Partition step.* Constructing and randomly partitioning the edge set $\{(v_i, v_j) : 1 \le i < j \le n\}$ into $K$ folds touches each edge once, so the cost is $O(n^2)$.

(2)  *Imputation step.* For a fixed fold $k$, constructing the training matrix $\mathbf{A}^{\text{train}}$ by masking and imputing the edges in $S_k$ visits $|S_k| \asymp n^2 / K$ entries, so the cost is $O(n^2 / K)$ per fold and $O(n^2)$ over all $K$ folds.

(3)  *Training step.* Fitting model $M$ to $\mathbf{A}^{\text{train}}$ has cost $C_{estim}(n)$ per fold, hence $O\bigl(K \cdot C_{estim}(n)\bigr)$ over all folds.

(4) *Debiasing step.* In fold $k$, the debiased predictions are computed only for the edges in validation set $S_k$, because these are the entries involved in the fold-$k$ validation loss. Since $|S_k| \asymp n^2/K$, the cost is $O(n^2/K)$ per fold and $O(n^2)$ overall across all folds.


(5) *Validation score.* For each fold $k$, the validation error is computed by comparing the held-out edges $a_{ij}$, $(i,j)\in S_k$, with their debiased predictions.  Since $|S_k|\asymp n^2/K$, the cost is $O(n^2/K)$ per fold and $O(n^2)$ across all folds.

Therefore, if we have $|\mathcal{M}|$ hyperparameters to evaluate,  the total cost of CV-imputation for hyperparameter tuning is $O(|\mathcal{M}| \cdot (KC_{estim}(n) + n^2))$.

ECV follows the same outer loop structure with $K$ folds and identical partitioning. The critical difference lies in step (2): while our method performs imputation only, ECV performs both imputation (in a different way, i.e., masking $S_k$ to all zero) and matrix completion on the masked $n  \times n$ matrix to produce a completed surrogate before estimation. This matrix completion step is computationally heavy, as it must operate on the full matrix and is repeated once per fold. The subsequent estimation and validation steps follow the same computational order as described above, except that ECV omits the de-biasing step. Therefore, ECV's total complexity is $O(|\mathcal{M}| \cdot   (KC_{estim}(n) + KT_{\text{mc}}(n)))$.

**Action:** (1) On page 4 in the revised manuscript, we have added a paragraph summarizing this computational complexity comparison. (2) On pages 16–18, we have added a new Section S.8 that provides the detailed breakdown of the computational complexity for CV-imputation and ECV.

### **2. Generalizability of CV-imputation to broader class of  models**


**Response:** We appreciate reviewers' insightful comment. Here, we present empirical evidence and discuss how our theoretical results can be extended to two representative settings: latent-space models and sparse network models.



#### **2.1 New Empirical Evidence**

(1) Latent space model. We first evaluated CV-imputation on latent space models where latent positions $\mu_i$ were sampled from uniform distribution over $[0,1]^d$, and edge probabilities were generated via an inner-product function $f(\mu_i, \mu_j) = (1+\mu_i^T \mu_j) / d$. We considered four settings: $d \in 2, 4, 6, 8$.

For each simulated network, we applied the NS and USVT estimation method to obtain estimates of the probability matrix. Although the NS and USVT methods were originally developed for graphon models, they remain applicable as general-purpose procedures that take the adjacency matrix as input and output an estimate of the edge probability matrix. In this experiment, our focus is on evaluating how effectively different hyperparameter selection strategies guide these estimators.

---

> ### Author Response · Authors · 2025-11-20
> **General Responses (2/3)**
>
> We compared four strategies: (i) CV-imputation, selecting the hyperparameter that minimizes our proposed cross-validation score; (ii) ECV, selecting the hyperparameter that minimizes the ECV score; (iii) Default, using a fixed, pre-determined hyperparameter value; and (iv) Oracle, using the hyperparameter that minimizes the Mean Squared Error (MSE) against the true probability matrix, representing the best achievable performance.
>
> Table R1 shows that CV-imputation is not only superior to existing data-driven tuning methods but is also capable of achieving performance  close to the oracle. This provides empirical evidence that the proposed method remains effective in latent space models.
>
> Table R1: Mean $\pm$  standard deviation of the MSE across 100 replicates (multiplied by 100 for readability), with $n=100$.
>
>  |             | $d=2$        | $d=4$          | $d=6$         | $d=8$         |
> |------------------------|------------------|------------------|------------------|------------------|
> | CV-imputation (NS)   | **1.12 (0.09)**    | **2.26 (0.13)**    | **2.19 (0.09)**| **1.67 (0.07)**|
> | ECV (NS)                   | 1.12 (0.17) | 3.04 (0.16)    | 2.19 (0.09)    | 1.83 (0.69)    |
> | Default NS (M = 1)         | 1.21 (0.08)    | 2.26 (0.13) | 3.10 (0.24)    | 3.33 (0.25)    |
> | Oracle (NS)         | 0.93 (0.10)  | 2.19 (0.12)  | 2.14 (0.09)  | 1.67 (0.07)  |
> | CV-imputation (USVT)   | **1.26 (0.10)**  | **1.99 (0.24)**  | **2.33 (0.16)**  | **1.88 (0.22)**  |
> | ECV (USVT)                   | 2.57 (0.73)      | 3.39 (1.13)      | 4.11 (0.79)      | 2.75 (0.31)      |
> | Default USVT (M = 0.01)         | 2.57 (0.73)      | 8.90 (0.26)      | 4.51 (0.12)      | 2.84 (0.08)      |
> | Oracle (USVT)          | 1.26 (0.10)  | 1.83 (0.16)  | 2.17 (0.12)  | 1.88 (0.12)  |
>
>
>
> (2) Sparsified Model: We simulated from a generalized sparse graphon model [1] defined by $p_{ij} = \rho_n \cdot f(\mu_i, \mu_j)$. Here, we used the same  $f(\cdot)$ as Graphons 1–4 in our main text (Figure 2), referring to their sparse counterparts as S1–S4. We introduced sparsity by setting the scaling factor $\rho_n = \sqrt{\frac{\log n}{n}}$. We tested this setting with $n=100$.  Table R2 confirms that CV-imputation generalizes effectively to sparse networks. It consistently outperforms ECV, achieving significantly lower MSEs that closely match the Oracle. This demonstrates that our method is a robust even when the dense network assumption is violated.
>
> Table R2: Mean $\pm$  standard deviation of the MSE across 100 replicates (multiplied by 100 for readability), with $n=100$.
>
> | Method                     | S1            | S2              | S3              | S4              |
> |----------------------------|---------------|------------------|------------------|------------------|
> | CV-imputation (NS)     | **0.48 (0.10)**| **3.07 (0.34)**  | **1.06 (0.11)**  | **1.04 (0.17)**      |
> | ECV (NS)               | 0.54 (0.11)    | 4.46 (0.49)      | 2.85 (1.22)      | 1.04 (0.16)  |
> | Default NS (M=1)       | 42.83 (6.58)   | 3.25 (0.25)      | 1.06 (0.12)      | 1.55 (0.23)      |
> | Oracle (NS)            | 0.45 (0.10)| 2.83 (0.22)  | 1.05 (0.12)  | 1.03 (0.16)  |
> |  CV-imputation (USVT)     | **0.43 (0.10)**  | **3.58 (0.24)**   | **1.19 (0.09)**   | **0.91 (0.13)**       |
> | ECV (USVT)                | 0.62 (0.10)      | 6.50 (1.72)       | 2.14 (1.31)       | 0.97 (0.37)   |
> | Default USVT (M = 0.01)       | 0.62 (0.10)      | 7.44 (0.40)       | 3.56 (1.73)       | 2.80 (0.66)       |
> | Oracle (USVT)           | 0.37 (0.06)  | 3.41 (0.21)   | 1.06 (0.17)   | 0.91 (0.13)   |
>
>
> #### **2.2 Theoretical Extension**
>
> **Extending Lemma 1.** Lemma 1 establishes independence between the training network and validation set conditional on $P$, requiring only conditional edge independence:  $a_{ij} \mid \mathbf{P} \sim \mathrm{Bernoulli}(p_{ij})$ are mutually independent given $\mathbf{P}$.
>
> This assumption is widely satisfied. In latent-space models,  [2] show edges are independent Bernoulli variables given latent positions. In sparse graphon models,  [1] demonstrate that edges are sampled independently with probabilities $\rho_n f(\mu_i,\mu_j)$ conditional on latent variables. Therefore, Lemma 1 directly extends to both settings.
>
>
>
> **Extending Theorem 1.** Theorem 1 requires two assumptions: (i) the conditional edge independence assumption, and  (ii) Condition 1. As discussed above, conditional edge independence holds for both latent-space models and  generalized sparse graphon models. Condition 1 requries  stability: masking and imputing a $1/K$ fraction of edges must not substantially perturb the fitted estimator. Since stability depends on the estimation method, Condition 1 must be verified for each specific estimation method.

---

> ### Author Response · Authors · 2025-11-20
> **General Responses (3/3)**
>
> ***Latent-space models.*** We verify Condition 1 for the Maximum Likelihood Estimator (MLE) in latent space models 3], where $a_{ij}\sim \mathrm{Bernoulli}(p_{ij})$ with $p_{ij} = \sigma(\mu_i^T\mu_j+\alpha_i+\alpha_j)$ with a fixed link function $\sigma(\cdot)$ and MLE used for estimation, where $\alpha_i, \alpha_j$ are degree parameters, $\sigma(\cdot)$ is a fixed link function.
>
> Let $\boldsymbol{\theta} = (\{\mu_i\}, \{\alpha_i\})$ denote the set of all latent parameters. Let $Q(\boldsymbol{\theta};\mathbf{A}) = \frac{2}{n(n-1)}\sum_{i<j} \ell(\boldsymbol{\theta}; \mathbf{A}) + R(\boldsymbol{\theta})$ denote the penalized log-likelihood using $\mathbf{A}$, where $l(\cdot)$ is the log-likelihood of bernoulli distribution, $R(\boldsymbol{\theta})$ is the penalty term. The score (i.e.,  gradient) is $\partial Q(\boldsymbol{\theta};\mathbf{A}) = \frac{2}{n(n-1)}\sum_{i<j} \partial \ell(\boldsymbol{\theta}; \mathbf{A}) + \partial R(\boldsymbol{\theta})$.
>
> **Score change under perturbation**. Under our CV-imputation procedure, the perturbation from $\mathbf{A}$ to  $\mathbf{A}^{\mathrm{train}}$ affects only entries in the validation set $S_k$, so only likelihood terms for node pairs in the validation set change. Thus,   $\Delta :=\partial Q(\boldsymbol{\theta};\mathbf{A}^{\mathrm{train}}) - \partial Q(\boldsymbol{\theta};\mathbf{A}) = \frac{2}{n(n-1)} \sum_{(i,j) \in S_k} \left[ \partial \ell(\boldsymbol{\theta}; \mathbf{A}^{\mathrm{train}}) - \partial \ell(\boldsymbol{\theta}; \mathbf{A}) \right]$.  Using Hölder's inequality, we obtain $\Vert \Delta\Vert_2^2 \leq \frac{2}{n(n-1)} \sum_{i,j \in S_k} \Vert\partial l(\sigma(\mu_i^T\mu_j+\alpha_i+\alpha_j) ;a_{ij}) - \partial l( \sigma(\mu_i^T\mu_j+\alpha_i+\alpha_j) ;b_{ij}) \Vert_2^2$,
>
> **Score change bound**. Using this inequation formula,  the central limit theorem and the regularization assumptions in [3], it is straightforward to prove that $\Vert \partial Q(\boldsymbol{\theta};\mathbf{A}^{\text{train}}) - \partial Q(\boldsymbol{\theta};\mathbf{A}) \Vert_2^2$ is bounded by $O_p(\frac{1}{K})$. Intuitively, this rate arises because the statistical noise introduced by imputation scales linearly with the number of perturbed terms ($|S_k| \asymp O(n^2/K)$). This $O(n^2/K)$ scaling is then divided by the squared number of observations ($O(n^2)$). The resulting cancellation leaves the perturbation rate proportional to $1/K$.
>
> **Estimator stability.** If the link function $\sigma(\cdot)$ is Lipschitz continuous (which is very easy to satisfy, e.g., $\sigma(x)=x$), the score  $\partial Q(\boldsymbol{\theta};\mathbf{A})$ can be shown to be Lipschitz continuous as well. Therefore, standard M-estimator theory indicates that this score stability translates directly to the parameter estimates, yielding that the $L_2-$distance between estimated probability matrices based on $\mathbf{A}$ and $\mathbf{A}^{\text{train}}$ is bounded by  $O_p(\frac{1}{K})$.
>
> **Conclusion**: For the Random Dot Product Graph (a special case of latent space model with $\sigma(x) = x$, $\alpha = 0_{n \times 1}$), the above derived bound implies the optimism bias $Q_K(M)$ is bounded by $O_p(\frac{1}{K^2})$. Thus, Condition 1 is satisfied with $\alpha=2$, and Theorem 1 holds.
>
>
> ***Generalized sparse graphon models.*** For sparse graphon model, $a_{ij}\sim \mathrm{Bernoulli}(\rho_n p_{ij})$ with the sparsity parameter $\rho_n \to 0$. To satisfy Condition 1, the training network must preserve the same sparsity order as the original network; otherwise, the estimator may converge at a different rate. As discussed in COROLLARY 3.3 of [4], estimating sparse graph with different sparsity levels using the block estimator yields different estimation error rates. Therefore, the choice of the imputation level  $\theta$ cannot be arbitrary. For example, setting $\theta=1$ could potentially create a dense training network and violate the sparsity assumptions required by graphon estimators designed for sparse graphs. Furthermore, extending Theorem 1 requires replacing the rate  $O_p(n^{-1})$ with the sparsity-adjusted rate typical in sparse graphon estimation, which involves additional $\rho_n$ factors.
>
> A full extension of Theorem 1 would therefore incorporate $\rho_n$ explicitly and verify that $Q_K(M)$ decays fast enough relative to these sparsity-adjusted rates. While our preliminary empirical results (Table R2) suggest that CV-imputation performs well in sparse settings, establishing formal guarantees remains an important direction for future work.
>
>
> **Action**: On page 10 in Section 7, we have (1) added discussions about the generalizability of our method to broader graph models, (2) added discussions about limitations, e.g., our theory is not applied to temporal networks since edges are dependent across time. Additionally, we incorporated new empirical results and discussion of potential theoretical extensions into Section S.9 of the Appendix.

---

> ### Author Response · Authors · 2025-11-20
> **References**
>
> References:
>
> [1] Klopp, O., Tsybakov, A. B., & Verzelen, N. (2017). Oracle inequalities for network models and sparse graphon estimation. Annals of Statistics, 45(1), 316-354.
>
> [2] Hoff, Peter D., Adrian E. Raftery, and Mark S. Handcock. "Latent space approaches to social network analysis." Journal of the American Statistical Association 97, no. 460 (2002): 1090-1098.
>
> [3] Li, Jinming, et al. "Statistical inference on latent space models for network data." arXiv preprint arXiv:2312.06605 (2023).
>
>
> [4] Athreya, Avanti, et al. "Statistical inference on random dot product graphs: a survey." Journal of Machine Learning Research 18.226 (2018): 1-92

---

### Meta-Review · Area_Chair_JziQ · 2026-01-04

**Summary:**

The reviewers have raised the following major concerns:

(1) The scope of the work requires further extension.

(2) There is a theoretical gap, and a more rigorous proof is needed.

(3) Scalability remains a concern.

(4) The empirical results lack sufficient in-depth analysis.

(5) The role of the graphon function requires further clarification.

**Reviewer Concerns:**

After reviewing the discussion-phase comments, I find that most of the reviewers’ concerns have been adequately addressed, and the overall evaluation of the submission is generally positive.

However, upon reading the paper myself, I have some reservations regarding Theorem 1. Specifically, in the left-hand-side expression, $V_K(M)$ denotes the prediction error (Eq. (7)), while $L(M)$ represents the (optimal) mean squared error (Eq. (3)). I initially expected the result to establish a bound on $|V_K(M) - L(M)|$. Instead, the bound includes an additional $\Lambda$ term, which may remain $O(1)$ even as $n \to \infty$. This suggests that the estimated error $V_K(M)$ could differ from $L(M)$ by an approximately constant amount, which, in my view, may be insufficient to fully justify the use of $V_K(M)$. Moreover, the discussion following Theorem 1 does not, at least to me, provide a fully convincing explanation of this point.

In summary, I believe the submission has the potential to be a meaningful contribution to ICLR. However, as Theorem 1 constitutes the central result of the paper, I would suggest that the ACs and the SAC discuss this issue before a final decision is made.

**Reviewer Scores:**

Reviewer 8g5b has indicated his/her willingness to increase the score.

---

### Decision · Program_Chairs · 2026-01-26

Accept (Poster)